# Protein prediction for trait mapping in diverse populations

**Ryan Schubert**[1,2,3], **Elyse Geoffroy**[3], **Isabelle Gregga**[2], **Ashley J. Mulford**[2,3], **Francois Aguet**[4], **Kristin Ardlie**[4], **Robert Gerszten**[5], **Clary Clish**[4], **David Van Den Berg**[6], **Kent D. Taylor**[7], **Peter Durda**[8], **W. Craig Johnson**[9], **Elaine Cornell**[8], **Xiuqing Guo**[7], **Yongmei Liu**[10], **Russell Tracy**[8], **Matthew Conomos**[11], **Tom Blackwell**[12], **George Papanicolaou**[13], **Tuuli Lappalainen**[14], **Anna V. Mikhaylova**[11], **Timothy A. Thornton**[11], **Michael H. Cho**[15], **Christopher R. Gignoux**[16], **Leslie Lange**[16], **Ethan Lange**[16], **Stephen S. Rich**[17], **Jerome I. Rotter**[7], **NHLBI TOPMed Consortium**[¶], **Ani Manichaikul**[17], **Hae Kyung Im**[18], **Heather E. Wheeler**[2,3] *

1 Department of Mathematics and Statistics, Loyola University Chicago, Chicago, IL, United States of America, 2 Department of Biology, Loyola University Chicago, Chicago, IL, United States of America, 3 Program in Bioinformatics, Loyola University Chicago, Chicago, IL, United States of America, 4 Broad Institute, Cambridge, MA, United States of America, 5 Beth Israel Deaconess Medical Center, Boston, MA, United States of America, 6 University of Southern California, Los Angeles, CA, United States of America, 7 The Institute for Translational Genomics and Population Sciences, Department of Pediatrics, The Lundquist Institute for Biomedical Innovation at Harbor-UCLA Medical Center, Torrance, CA, United States of America, 8 Laboratory for Clinical Biochemistry Research, University of Vermont, Burlington, VT, United States of America, 9 Collaborative Health Studies Coordinating Center, University of Washington, Seattle, WA, United States of America, 10 Department of Medicine, Duke University School of Medicine, Durham, NC, United States of America, 11 Department of Biostatistics, University of Washington, Seattle, WA, United States of America, 12 Department of Biostatistics, University of Michigan, Ann Arbor, MI, United States of America, 13 Epidemiology Branch, National Heart, Lung and Blood Institute, Bethesda, MD, United States of America, 14 New York Genome Center and Department of Systems Biology, Columbia University, New York, NY United States of America, 15 Channing Division of Network Medicine, Brigham and Women's Hospital, Boston, MA, United States of America, 16 Division of Biomedical Informatics and Personalized Medicine, Department of Medicine, University of Colorado Anschutz Medical Campus, Aurora, CO, United States of America, 17 Center for Public Health Genomics, University of Virginia, Charlottesville, VA, United States of America, 18 Section of Genetic Medicine, The University of Chicago, Chicago, IL, United States of America

¶ Membership list can be found in the S8 Table.
* hwheeler1@luc.edu

**Data Availability Statement:** Models presented in the main text are available at https://doi.org/10.5281/zenodo.4837327. Code for supporting figures and analysis are available at

## Abstract

Genetically regulated gene expression has helped elucidate the biological mechanisms underlying complex traits. Improved high-throughput technology allows similar interrogation of the genetically regulated proteome for understanding complex trait mechanisms. Here, we used the Trans-omics for Precision Medicine (TOPMed) Multi-omics pilot study, which comprises data from Multi-Ethnic Study of Atherosclerosis (MESA), to optimize genetic predictors of the plasma proteome for genetically regulated proteome-wide association studies (PWAS) in diverse populations. We built predictive models for protein abundances using data collected in TOPMed MESA, for which we have measured 1,305 proteins by a SOMAscan assay. We compared predictive models built via elastic net regression to models integrating posterior inclusion probabilities estimated by fine-mapping SNPs prior to elastic net. In order to investigate the transferability of predictive models across ancestries, we built protein prediction models in all four of the TOPMed MESA populations, African American (n = 183), Chinese (n = 71), European (n = 416), and Hispanic/Latino (n = 301), as well as in all

https://github.com/RyanSchu/TOPMed_protein_
prediction. UKB GWAS summary statistics can be
accessed via http://www.nealelab.is/uk-biobank/.
All other large European GWAS can be accessed
through the GWAS catalog. A list of studies can be
found in S2 Table. Data from INTERVAL is under
controlled access via the European Genome-
phenome Archive hhttps://ega-archive.org/ for both
genotypes (EGAD00010001544) and 638 blood
plasma aptamers levels as measured by a
SOMAscan assay (EGAD00001004080). PAGE
GWAS summary statistics are available in the
GWAS Catalog at https://www.ebi.ac.uk/gwas/
publications/31217584. MESA data are under
controlled access in dbGaP at https://www.ncbi.
nlm.nih.gov/gap/. Genotypes are available through
accession phs000420.v6.p3 and protein data will
be available through accession phs001416.v2.p1.

**Funding:** This work is supported by the NIH
National Human Genome Research Institute
Academic Research Enhancement Award R15
HG009569 (HEW). MESA and the MESA SHARe
project are conducted and supported by the
National Heart, Lung, and Blood Institute (NHLBI)
in collaboration with MESA investigators. Support
for MESA is provided by contracts
75N92020D00001, HHSN268201500003I, N01-
HC-95159, 75N92020D00005, N01-HC-95160,
75N92020D00002, N01-HC-95161,
75N92020D00003, N01-HC-95162,
75N92020D00006, N01-HC-95163,
75N92020D00004, N01-HC-95164,
75N92020D00007, N01-HC-95165, N01-HC-
95166, N01-HC-95167, N01-HC-95168, N01-HC-
95169, UL1-TR-000040, UL1-TR-001079, UL1-
TR-001420. Funding for SHARe genotyping was
provided by NHLBI Contract N02-HL-64278.
Genotyping was performed at Affymetrix (Santa
Clara, California, USA) and the Broad Institute of
Harvard and MIT (Boston, Massachusetts, USA)
using the Affymetrix Genome-Wide Human SNP
Array 6.0. MESA Family is conducted and
supported by the National Heart, Lung, and Blood
Institute (NHLBI) in collaboration with MESA
investigators. Support is provided by grants and
contracts R01HL071051, R01HL071205,
R01HL071250, R01HL071251, R01HL071258,
R01HL071259, by the National Center for Research
Resources, Grant UL1RR033176. Also supported
in part by the National Center for Advancing
Translational Sciences, CTSI grant UL1TR001881,
and the National Institute of Diabetes and Digestive
and Kidney Disease Diabetes Research Center
(DRC) grant DK063491 to the Southern California
Diabetes Endocrinology Research Center. The
TOPMed MESA Multi-Omics project was
conducted by the University of Washington and

populations combined. As expected, fine-mapping produced more significant protein predic-
tion models, especially in African ancestries populations, potentially increasing opportunity
for discovery. When we tested our TOPMed MESA models in the independent European
INTERVAL study, fine-mapping improved cross-ancestries prediction for some proteins.
Using GWAS summary statistics from the Population Architecture using Genomics and Epi-
demiology (PAGE) study, which comprises ∼50,000 Hispanic/Latinos, African Americans,
Asians, Native Hawaiians, and Native Americans, we applied S-PrediXcan to perform
PWAS for 28 complex traits. The most protein-trait associations were discovered, coloca-
lized, and replicated in large independent GWAS using proteome prediction model training
populations with similar ancestries to PAGE. At current training population sample sizes,
performance between baseline and fine-mapped protein prediction models in PWAS was
similar, highlighting the utility of elastic net. Our predictive models in diverse populations are
publicly available for use in proteome mapping methods at https://doi.org/10.5281/zenodo.
4837327.

## Introduction

Genome-wide association studies (GWAS) have uncovered novel genetic associations under-
pinning a wide array of complex traits [1–10]. Methods like PrediXcan and FUSION have suc-
cessfully integrated underlying gene regulation mechanisms in gene mapping studies [11, 12].
In these so-called transcriptome-wide association studies (TWAS), reference expression quan-
titative trait loci (eQTL) data are used to build models that predict gene expression levels from
genotypes. The models are integrated with GWAS data to test genes, rather than SNPs, for
association with complex traits. TWAS have a lower multiple testing correction burden than
GWAS and provide clear gene targets for future investigations [13, 14]. In addition, TWAS
inherently include information such as direction of effect for a gene on a trait that is not often
apparent at the SNP level.

Like polygenic risk scores, the efficacy of predictive models at the transcriptome level is
reduced by differences in linkage disequilibrium (LD), allele frequencies, and effect sizes across
populations [15–20]. The exclusion of non-European ancestry populations from much of
human genetics diminishes the promise of precision medicine and misses opportunities for
fine-mapping and locus discovery [21, 22]. Population-matched transcriptome prediction
increases TWAS discovery and replication rate [23]. Thus, as multi-omics studies increase and
methods like PrediXcan expand to include omics traits beyond the transcriptome, inclusion of
diverse ancestral populations is crucial. With the advent of high-throughput proteome tech-
nologies [24, 25], many studies have identified protein QTLs (pQTLs), especially in plasma
and European ancestries populations [26–28]. Like eQTLs, GWAS are often enriched in
pQTLs, and proteome-wide association studies (PWAS) have been proposed [29, 30].

Here, we used the TOPMed Multi-omics pilot study [25], which comprises data from
MESA [31], to optimize genetic predictors of the plasma proteome for PWAS. We trained pro-
tein prediction models using genotype and plasma proteome data from an aptamer-based
assay of 1305 proteins from 971 individuals of African American, Chinese, European, and His-
panic/Latino populations. We compared model building methods that included fine-mapping
to baseline elastic net within each population and across all populations. We tested our protein
prediction models in the independent INTERVAL study [26] and show that while fine-

LABioMed (HHSN2682015000031/
HHSN26800004). Molecular data for the Trans-
Omics in Precision Medicine (TOPMed) program
was supported by the National Heart, Lung and
Blood Institute (NHLBI). SOMAscan proteomics for
NHLBI TOPMed: Multi-Ethnic Study of
Atherosclerosis (MESA) (phs001416.v1.p1) was
performed at the Broad Institute and Beth Israel
Proteomics Platform (HHSN268201600034I). Core
support including centralized genomic read
mapping and genotype calling, along with variant
quality metrics and filtering were provided by the
TOPMed Informatics Research Center (3R01HL-
117626-02S1; contract HHSN268201800002I).
Core support including phenotype harmonization,
data management, sample-identity QC, and general
program coordination were provided by the
TOPMed Data Coordinating Center (R01HL-
120393; U01HL-120393; contract
HHSN268201800001I). We gratefully acknowledge
the studies and participants who provided
biological samples and data for TOPMed.
Participants in the INTERVAL randomised
controlled trial were recruited with the active
collaboration of NHS Blood and Transplant England
(www.nhsbt.nhs.uk), which has supported field
work and other elements of the trial. DNA
extraction and genotyping was co-funded by the
National Institute for Health Research (NIHR), the
NIHR BioResource (http://bioresource.nihr.ac.uk)
and the NIHR [Cambridge Biomedical Research
Centre at the Cambridge University Hospitals NHS
Foundation Trust]. The INTERVAL study was
funded by NHSBT (11-01-GEN). The academic
coordinating centre for INTERVAL was supported
by core funding from: NIHR Blood and Transplant
Research Unit in Donor Health and Genomics
(NIHR BTRU-2014-10024), UK Medical Research
Council (MR/L003120/1), British Heart Foundation
(SP/09/002; RG/13/13/30194; RG/18/13/33946)
and the NIHR [Cambridge Biomedical Research
Centre at the Cambridge University Hospitals NHS
Foundation Trust]. Proteomic assays were funded
by the academic coordinating centre for INTERVAL
and MRL, Merck & Co., Inc. A complete list of the
investigators and contributors to the INTERVAL
trial is provided in Di Angelantonio et al. The
academic coordinating centre would like to thank
blood donor centre staff and blood donors for
participating in the INTERVAL trial. This work was
supported by Health Data Research UK, which is
funded by the UK Medical Research Council,
Engineering and Physical Sciences Research
Council, Economic and Social Research Council,
Department of Health and Social Care (England),
Chief Scientist Office of the Scottish Government
Health and Social Care Directorates, Health and

mapping may improve cross-population prediction performance, larger sample sizes are needed to increase confidence in independent signals. We also applied S-PrediXcan [32] to the PAGE Study GWAS summary statistics [1] to assess model performance in a PWAS framework. PrediXcan [11] requires genotype data to estimate expression levels for use in association testing, but S-PrediXcan [32] requires only GWAS summary statistics to perform TWAS. The LD reference information for S-PrediXcan comes from the protein prediction model training population. We show population-matched protein prediction models yield more reliable associations, defined by colocalization and independent replication in large European GWAS, including those available from UKBiobank. We make all protein prediction models publicly available at https://doi.org/10.5281/zenodo.4837327

## Results

### Fine-mapping integration in protein abundance prediction model training

We set out to provide a useful resource for proteome association discovery in diverse populations. We first performed cis-pQTL mapping in each each TOPMed MESA population, which included African Americans (AFA, n = 183), Chinese (CHN, n = 71), Europeans (EUR, n = 416), Hispanic/Latinos (HIS, n = 301), and all populations combined (ALL, n = 971) (S1 Fig). We tested SNPs within 1 Mb of the gene for association with protein aptamer levels. Increasing sample size corresponded to more pQTL associations found in TOPMed MESA (FDR < 0.05, Table 1). Relative to eQTL studies, we found fewer pQTLs because of the smaller set of proteins (1305) that were available to test. Cis-pQTL summary statistics are available at https://doi.org/10.5281/zenodo.4837327. We found that effect sizes were enriched near the transcription start site (TSS) for each gene region which mapped to a protein in our sample and that as sample size increased, smaller effect size SNP associations farther from the TSS were discovered (S2 Fig).

We sought a balance between protein prediction model performance and maximizing the number of proteins that can be tested for association with complex traits in PWAS. We compared baseline and fine-mapped elastic net models predicting protein levels from SNP genotypes in each TOPMed MESA population. We used the effect sizes generated in our cis-pQTL analyses in the fine mapping. Using the same thresholds for significance as PrediXcan transcriptome modeling [11, 33], we quantified model quality by counting the number of protein models with cross validated $\rho > 0.1$ and $p < 0.05$ within each population and model building strategy.

We tested several posterior inclusion probability (PIP) thresholds and LD cluster filtering decisions to optimize our fine-mapping strategy (S1 Table). At all thresholds, our

**Table 1. pQTL counts (FDR <0.05) in TOPMed MESA populations.** Some proteins had more than one aptamer targeting it.

| Population | number pSNPs | number pAptamers | number pGenes |
|---|---|---|---|
| CHN (n = 71) | 2,990 | 73 | 72 |
| AFA (n = 183) | 4,564 | 153 | 150 |
| HIS (n = 301) | 16,164 | 273 | 270 |
| EUR (n = 416) | 30,761 | 260 | 253 |
| ALL (n = 971) | 36,487 | 384 | 378 |

CHN = Chinese, AFA = African American, HIS = Hispanic/Latino, EUR = European, ALL = All TOPMed MESA-defined groups combined.

Social Care Research and Development Division (Welsh Government), Public Health Agency (Northern Ireland), British Heart Foundation and Wellcome. The views expressed are those of the authors and not necessarily those of the NHS, the NIHR or the Department of Health and Social Care. The funders had no role in study design, data collection and analysis, decision to publish, or preparation of the manuscript.

**Competing interests:** The authors have declared that no competing interests exist.

fine-mapping strategy produced more predictive models compared to baseline, which we expected because we performed SNP-level fine-mapping in the full data set prior to cross-validated elastic net modeling (Fig 1, S3 Fig). Because all fine-mapped models within a population showed similar and higher correlation to each other than to baseline (S4 Fig), we chose to focus on one set of fine-mapped models, those with PIP > 0.001 and filtered LD clusters, to compare with baseline elastic net for the rest of the main text. The PIP > 0.001 and filtered LD clusters models, which we will now refer to as our "fine-mapped" models (Fig 1), balance performance with the number of proteins available for PWAS.

We found that 1187 unique protein aptamers have a significant prediction model across all training populations and both our baseline and fine-mapped model building strategies. While the smallest training population, CHN, produced the smallest number of models for either strategy, AFA, HIS, and EUR produce comparable numbers of models in spite of sample size differences (Fig 1B). For example, despite being less than half the size of the EUR population, about the same number of fine-mapped protein models were significant in AFA. This is likely due to more SNP variation in African ancestry populations, which leads to more features for prediction.

While the ALL combined population produced the most significant protein models in our baseline strategy, fine-mapping in ALL led to fewer protein models than in AFA, HIS, or EUR (Fig 1B). Fine-mapping in ALL may home in on cross-population associated variants with similar effect sizes at the expense of population-specific variation.

In addition, we determined if any of our significant protein models represented new genes not covered in previous transcriptome prediction modeling. As proteins measured in blood plasma may contain proteins excreted by a number of tissues, we compared our protein models to RNA models built in both Whole Blood as well as all 49 GTEx tissues [33]. In total, between both model building strategies and all training populations, we found 372 distinct protein aptamers with at least one predictive model that do not have an RNA equivalent model from GTEx v8 MASHR Whole Blood models, 18 of which do not have an RNA equivalent model in any tissue in GTEx v8 MASHR models [33] (S3 Table).

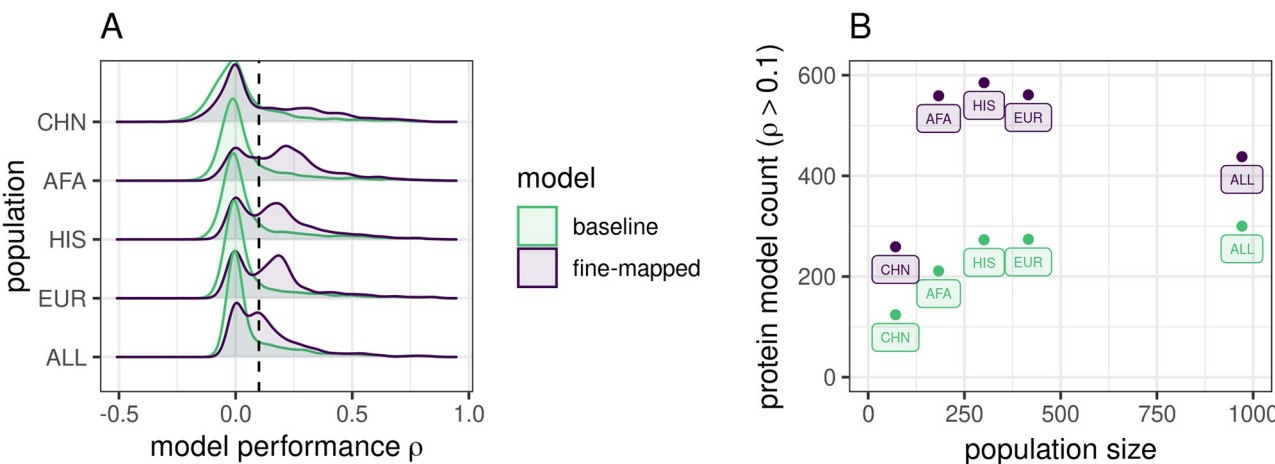

**Fig 1. Protein prediction performance in TOPMed MESA populations. A**. Distributions of prediction performance across proteins within each training population between modeling strategies. $\rho$ is the Spearman correlation between predicted and observed protein abundance in the cross-validation. Fine-mapping prior to elastic net modeling produces more significant ($\rho > 0.1$, vertical dotted line) protein prediction models than baseline elastic net. **B**. Significant ($\rho > 0.1$, $p < 0.05$) protein model counts compared to population sample size colored by modeling strategy. TOPMed MESA populations: CHN, Chinese; AFA, African American; HIS, Hispanic/Latino; EUR, European; ALL, all populations combined.

## Fine-mapping can improve cross-population protein prediction performance

While fine-mapping leads to more models which may allow for more associations to be discovered in PWAS, our strategy could lead to overfitting. Thus, we next assessed model performance by testing our TOPMed MESA models in an independent proteome study. We tested the performance of models trained in the TOPMed MESA populations for predicting protein levels from individual level genotypes using the INTERVAL study (n = 3301 Europeans) [26, 34]. We predicted protein abundance in INTERVAL using both fine-mapped and baseline models trained in each TOPMed MESA population, for a total of 10 model sets. Of the 804 protein aptamers measured within INTERVAL that map uniquely to the same aptamer measured in TOPMed MESA, 597 unique protein aptamers had a significant prediction model in at least one model set. As the heritability of a trait determines the ceiling for genetic prediction performance, we estimated the proportion variance explained (PVE) by SNPs within 1Mb of each protein encoding gene using Basyesian Sparse Linear Mixed Modeling (BSLMM) [35]. Highly heritable proteins (high PVE) were associated with high predictive performance in INTERVAL across populations, despite larger credible sets surrounding the PVE estimates in the smaller populations. (S5 Fig).

We compared the performance of the fine-mapped model set to baseline model set within each training population by comparing the distributions of the Spearman correlations using Wilcoxon signed-rank tests. Fine-mapped models trained in AFA and CHN had significantly better prediction in INTERVAL than baseline elastic net models, fine-mapped models trained in EUR and HIS were not significantly different, while fine-mapped models trained in ALL were significantly worse (Fig 2). Over the range of fine-mapping thresholds we tested, we found similar results. Fine-mapped models in AFA consistently outperformed baseline models, fine-mapped CHN was either significantly better or not different, and fine-mapped ALL, HIS, and EUR were either significantly worse or not different from baseline (S4 Table, S6 Fig).

Within each model building strategy, we were interested in comparing protein prediction performance in INTERVAL between the similar ancestries EUR training population and the larger, multi-ancestries ALL population. In order for a protein to be predicted in INTERVAL, at least one SNP in the MESA model must be polymorphic (MAF >0.01) in INTERVAL. Within the baseline models, more proteins were predicted in INTERVAL using the ALL training population (n = 183) compared to EUR (n = 149), with 107 shared proteins. However, more proteins were predicted with EUR fine-mapped models (n = 340) compared to ALL fine-mapped models (n = 259), with 183 shared proteins. Yet, for the proteins predicted by both training populations in INTERVAL, the ALL population predicted better with both the baseline (Wilcoxon signed-rank test $p = 0.0012$) and fine-mapped (Wilcoxon signed-rank test $p = 0.0064$) model building strategies (Fig 3). The mean difference of ALL—EUR prediction performance was larger, but with more variance, using the fine-mapped (mean [95% CI] = 0.018 [0.00070–0.036]) compared to baseline (mean [95% CI] = 0.0074 [0.0027–0.012]) models. Thus, fine-mapping across ancestries can be beneficial to prediction (Fig 3B).

When we compared all five TOPMed MESA training populations within each model building strategy, we observed the largest and most significant differences between populations in the baseline models rather than the fine-mapped models (S7 Fig, S5 and S6 Tables). To test the hypothesis that allele frequency differences between populations influence predictive power, we performed a fixation index ($F_{ST}$) analysis. For each model set, we calculated the ($F_{ST}$) between INTERVAL and the corresponding TOPMed population for SNPs in the predictive model. We then compared the difference in average ($F_{ST}$) between protein models that had a large difference in predictive performance between populations and protein models that had a

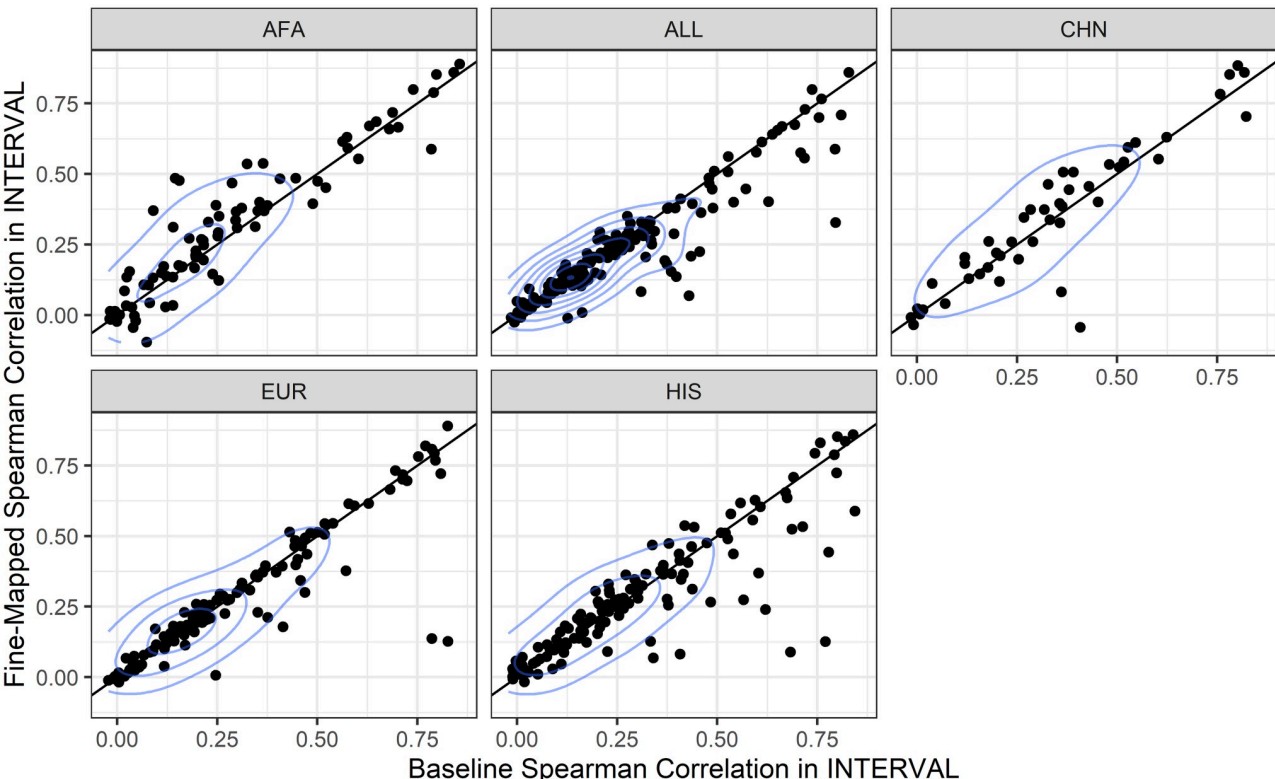

**Fig 2. TOPMed MESA protein prediction model performance comparison in the independent INTERVAL population.** Within each training population, the fine-mapped model performance in INTERVAL (y-axis) is compared to the baseline elastic net model performance in INTERVAL (x-axis). Each dot represents a protein that is predicted by both baseline models and fine-mapped models. Performance was measured as the Spearman $\rho$ between the measured protein aptamer level and the predicted protein aptamer level. Fine-mapped models performed better than baseline models in AFA (Wilcoxon signed-rank test, $p = 0.0016$) and CHN ($p = 0.036$), were not significantly different in EUR ($p = 0.74$) and HIS ($p = 0.54$), and significantly worse in ALL ($p = 0.0085$). TOPMed MESA populations: AFA, African American; ALL, all populations combined; CHN, Chinese; EUR, European; HIS, Hispanic/Latino.

small difference (Fig 4). We tested multiple thresholds for differences in predictive performance in both fine-mapped and baseline model sets. We found that models which had minimal differences in their performance had significantly smaller differences in average ($F_{ST}$) than models which had larger differences in performance by Wilcoxon signed-rank test (Fig 4). This effect was observed for multiple difference thresholds tested in both baseline and fine-mapped model sets, but was attenuated in fine-mapped sets. Thus, performance differences between populations in the fine-mapped models are less likely due to allele frequency differences. As sample sizes in proteomics studies increase, allowing identification of SNPs with higher PIP values, including trans-acting pQTLs, we anticipate increased cross-population performance benefit from multi-ancestries fine-mapping.

## Population-matched protein prediction models map the most trait associations

To test whether fine-mapping prior to model building leads to discovery of more protein-trait associations, we applied S-PrediXcan [32] using our TOPMed MESA prediction models to test proteins for association with the 28 phenotypes analyzed in the PAGE GWAS [1, 36]. Individuals in the PAGE study self-identified as Hispanic/Latino (n = 22,216), African American (n = 17,299), Asian (n = 4,680), Native Hawaiian (n = 3,940), Native American (n = 652), or

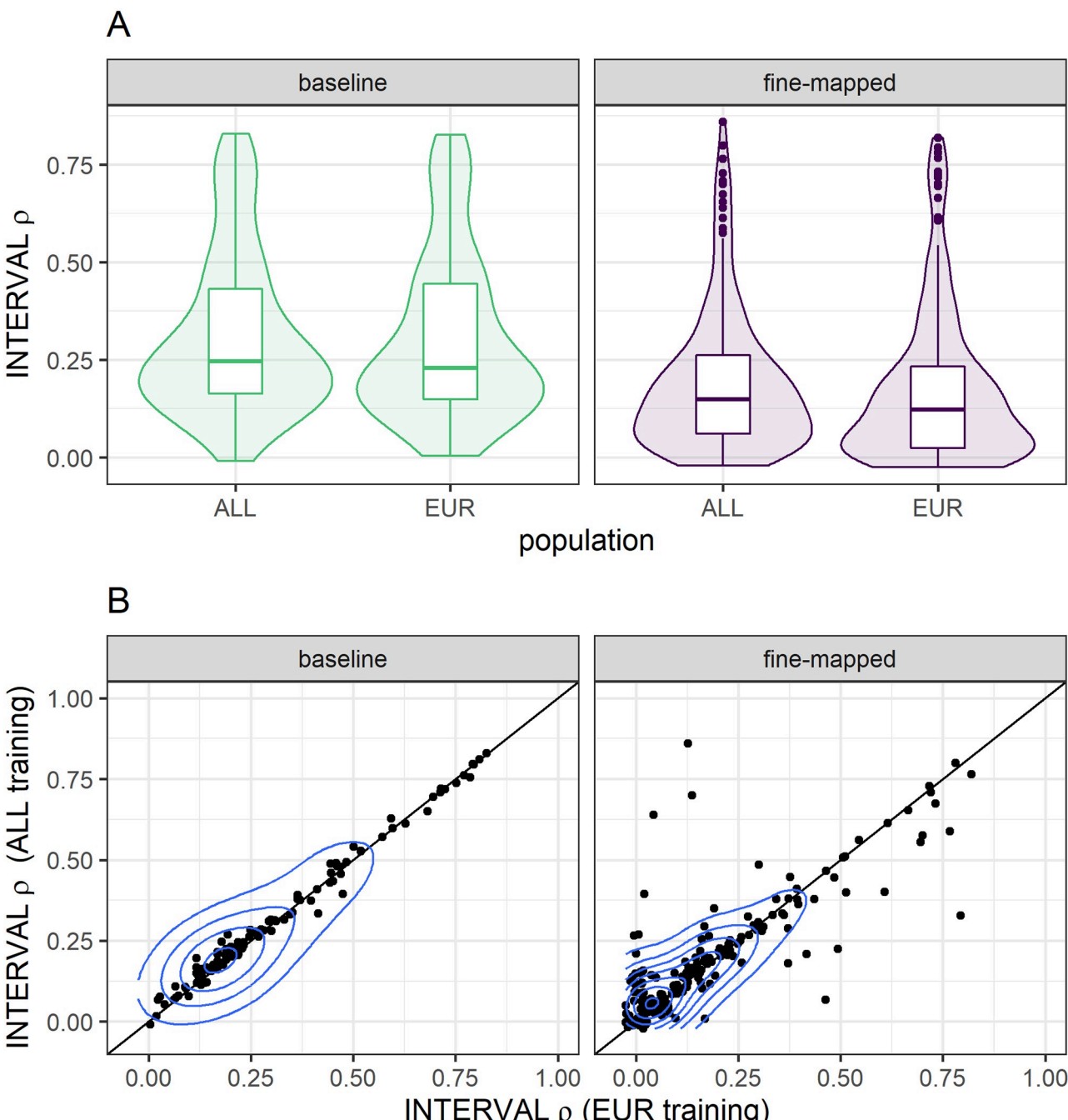

**Fig 3. Protein prediction performance between training populations within each model building strategy.** We compare the performance of TOPMed MESA ALL and EUR training populations in the INTERVAL study, a European population. For each model building strategy we first take the intersection of proteins that are predicted by both training populations and then test for differences in the distributions of Spearman correlation ($\rho$) by a Wilcoxon signed-rank test. INTERVAL $\rho$ was significantly higher when we used the ALL training population in both our baseline ($p = 0.0012$) and fine-mapped ($p = 0.0064$) modeling strategies. (A) The distributions of INTERVAL $\rho$ are plotted in each training population and modeling strategy. (B) The pairwise performance comparisons between ALL and EUR training populations are shown, each point represents a protein. The blue contour lines from two-dimensional kernel density estimation help visualize where the points are concentrated.

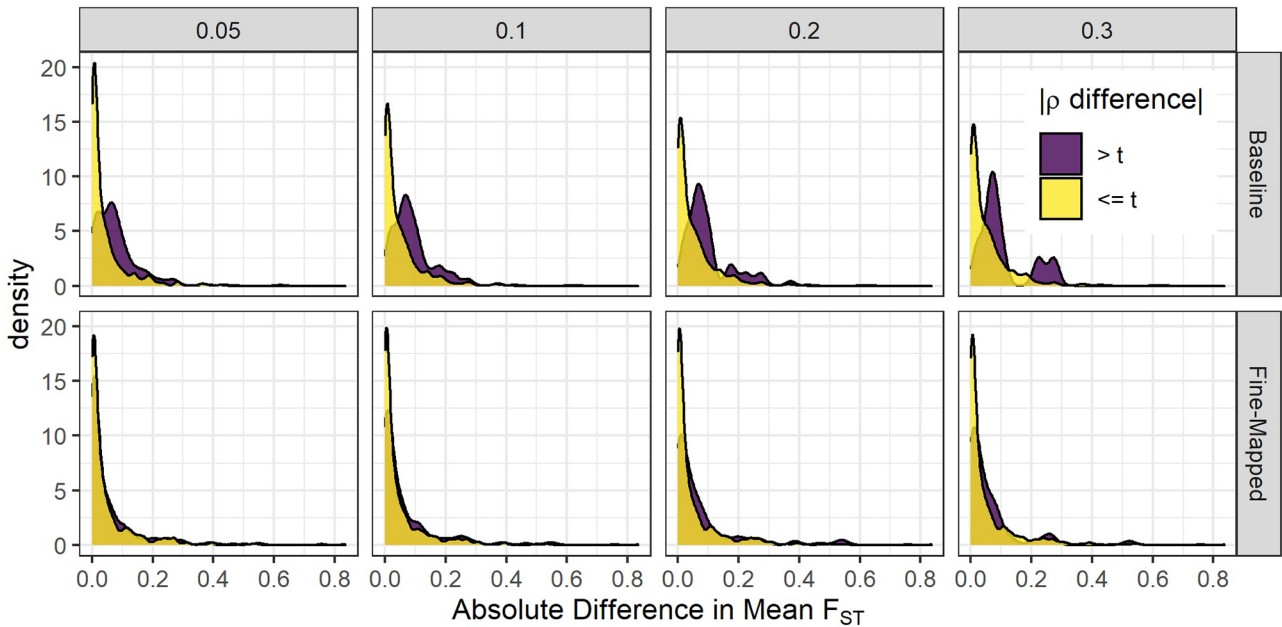

**Fig 4. Allele frequency differences lead to protein predictive performance differences between populations.** Comparison of mean $F_{ST}$ differences between protein models with large ($>t$) and small ($\le t$) differences in predictive performance $\rho$ in INTERVAL. For baseline models, protein groups with the larger absolute value $\rho$ difference between TOPMed MESA training populations had significantly larger mean $F_{ST}$ at each difference threshold, t (Wilcoxon rank sum tests, $p < 3.1 \times 10^{-10}$). For fine-mapped models, the differences between protein groups were attenuated, but still significant when t = 0.1 ($p = 0.0028$) and t = 0.2 ($p = 0.010$).

Other (n = 1,052) [1]. We identified a total of 29 distinct Bonferroni significant protein-trait associations using baseline elastic net models and 54 using fine-mapped models ($p < 1.54 \times 10^{-6}$ for baseline, $p < 7.60 \times 10^{-7}$ for fine-mapped, S7 Table). The most associations were found when applying models built in TOPMed AFA followed by TOPMed HIS, regardless of model building strategy (Fig 5A). We observed similar patterns for most fine-mapping thresholds tested (S8 Fig).

For protein-trait pairs discovered via S-PrediXcan, we then performed colocalization analysis to provide more evidence the SNPs in the protein region are acting through protein regulation to affect the associated phenotype. Similar numbers of distinct protein-trait associations are both S-PrediXcan significant and colocalized between baseline elastic net models (22) and fine-mapped models (21) (Fig 5B, S7 Table).

We then use the UKB+ GWAS summary statistics (see Methods) to survey which protein-trait pairs replicate in independent data. The majority of associations that are both colocalized and S-PrediXcan significant in PAGE replicated with the same direction of effect in the UKB+ data ($p < 1.54 \times 10^{-6}$ for baseline, $p < 9.59 \times 10^{-7}$ for fine-mapped; Fig 5C). Baseline elastic net models have the greatest number of protein-trait pairs which meet all three significance criteria (21) compared to fine-mapped models (17). Models trained in HIS and AFA have the most associations meeting all three significance criteria compared to the other training populations, likely reflective of the similar ancestries between AFA, HIS, and PAGE. Fine-mapped models trained in TOPMed HIS and TOPMed AFA generally have more protein-trait discoveries and replications compared to other training populations across PIP thresholds and clustering strategies (S8 Fig). In total we find 21 protein-trait associations that meet all three significance criteria (Table 2, S7 Table). Even though fine-mapping produced more models to test, a higher proportion of significant baseline-modeled proteins have colocalized SNP signals

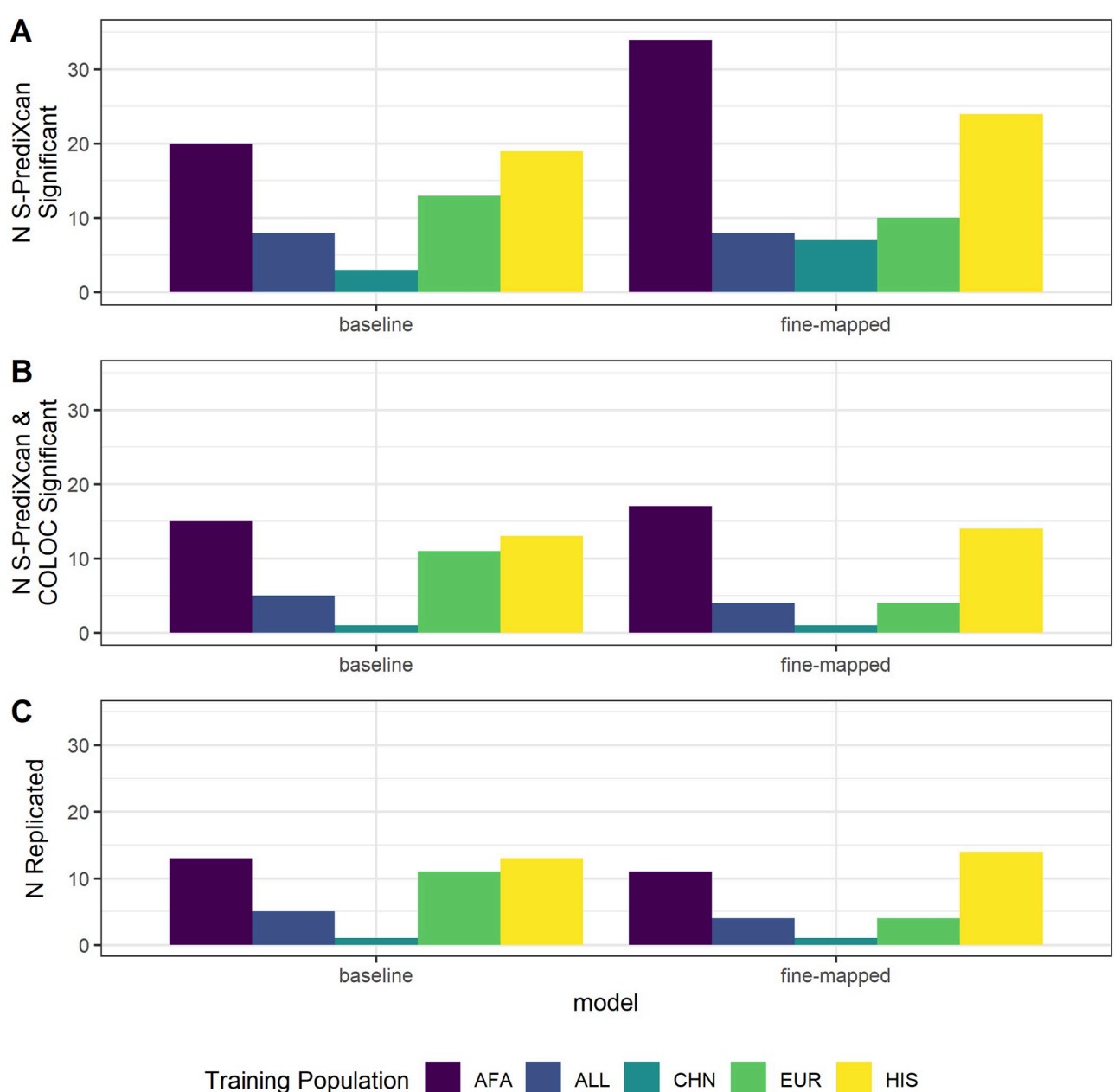

**Fig 5. Predicted protein-trait association results summary.** (A) Bonferroni significant (baseline $p < 1.54 \times 10^{-6}$; fine-mapped $p < 7.60 \times 10^{-7}$) protein-trait association counts when we applied S-PrediXcan to 28 traits in PAGE using protein prediction models from each TOPMed MESA population and model building strategy. (B) Protein-trait pairs from A that also have a COLOC colocalization probability > 0.5. (C) Protein-trait pairs from B that replicate (baseline $p < 1.54 \times 10^{-6}$; fine-mapped $p < 9.59 \times 10^{-7}$) in independent studies from the UKBioBank or other large, European ancestries cohorts. Bonferroni threshold for fine-mapped models is calculated separately from the Bonferroni threshold for baseline models.

between protein abundance and traits, with similar numbers of protein-trait associations that replicate in UKB+ studies between fine-mapped and baseline models ([Fig 5]).

We identified 21 distinct protein-phenotype associations which are Bonferroni significant in PAGE, colocalize in PAGE, and replicate with the same direction of effect in UKB+. These associations comprise eight distinct protein targets: total Apolipoprotein E and its three

**Table 2. Significant protein-trait associations found in PAGE, colocalized, and replicated in UKB+.** Each protein-phenotype pair may be present across multiple populations for different model building strategies. For each distinct protein-phenotype pair we present only the model association with the lowest p value in PAGE. All significant associations are listed in S7 Table.

| Aptamer | Protein | Phenotype | Train Pop | Model | PAGE $\beta$ | PAGE $p$ | UKB+ $\beta$ | UKB+ $p$ | PAGE coloc prob |
|---|---|---|---|---|---|---|---|---|---|
| SL000276* | Apo E | LDL cholesterol | AFA | Fine-Mapped | 15.65 | 4.22e-218 | 0.381 | 1.00e-51 | 0.991 |
| SL004668* | Apo E3 | LDL cholesterol | AFA | Fine-Mapped | 16.11 | 2.42e-217 | 0.396 | 1.00e-51 | 0.993 |
| SL000277* | Apo E2 | LDL cholesterol | HIS | Fine-Mapped | 19.44 | 7.77e-206 | 0.487 | 9.35e-57 | 0.991 |
| SL004669* | Apo E4 | LDL cholesterol | HIS | Fine-Mapped | 23.64 | 7.77e-206 | 0.593 | 9.35e-57 | 0.954 |
| SL000051 | CRP | C-reactive protein | ALL | baseline | 1.40 | 1.41e-122 | 1.03 | 3.05e-176 | 0.989 |
| SL000276* | Apo E | Total cholesterol | AFA | Fine-Mapped | 12.49 | 1.77e-114 | 0.290 | 1.00e-51 | 0.992 |
| SL000277* | Apo E2 | Total cholesterol | HIS | Fine-Mapped | 15.77 | 4.64e-111 | 0.371 | 1.00e-51 | 0.991 |
| SL004668* | Apo E3 | Total cholesterol | HIS | Fine-Mapped | 17.45 | 4.64e-111 | 0.411 | 1.00e-51 | 0.989 |
| SL004669* | Apo E4 | Total cholesterol | HIS | Fine-Mapped | 19.17 | 4.64e-111 | 0.451 | 1.00e-51 | 0.950 |
| SL001943 | IL-6 sRa | C-reactive protein | HIS | baseline | -0.121 | 1.51e-33 | -0.107 | 2.23e-308 | 0.996 |
| SL000277* | Apo E2 | C-reactive protein | EUR | baseline | -0.356 | 4.89e-27 | -0.466 | 1.82e-267 | 0.993 |
| SL004669* | Apo E4 | C-reactive protein | EUR | Fine-Mapped | -0.301 | 1.06e-26 | -0.313 | 5.68e-73 | 0.991 |
| SL004669* | Apo E4 | HDL cholesterol | HIS | baseline | -6.60 | 4.15e-25 | -0.184 | 6.18e-56 | 0.950 |
| SL000277* | Apo E2 | HDL cholesterol | HIS | Fine-Mapped | -2.37 | 7.29e-25 | -0.070 | 4.25e-59 | 0.991 |
| SL000276* | Apo E | HDL cholesterol | HIS | Fine-Mapped | -2.25 | 7.29e-25 | -0.066 | 4.25e-59 | 0.996 |
| SL004668* | Apo E3 | HDL cholesterol | HIS | Fine-Mapped | -2.62 | 7.29e-25 | -0.077 | 4.25e-59 | 0.989 |
| SL000276* | Apo E | C-reactive protein | EUR | baseline | -0.223 | 1.37e-13 | -0.310 | 9.46e-176 | 0.993 |
| SL004668* | Apo E3 | C-reactive protein | EUR | baseline | -0.235 | 1.28e-12 | -0.361 | 5.25e-161 | 0.985 |
| SL001990 | IL-1Ra | C-reactive protein | ALL | baseline | -0.188 | 1.30e-10 | -0.136 | 5.01e-65 | 0.981 |
| SL000437 | Haptoglobin, Mixed Type | LDL cholesterol | ALL | baseline | -1.86 | 1.11e-9 | -0.051 | 2.03e-114 | 0.985 |
| SL000437 | Haptoglobin, Mixed Type | Total cholesterol | ALL | baseline | -2.07 | 1.79e-9 | -0.048 | 1.90e-105 | 0.984 |

*Association is no longer significant after PAV adjustment.

AFA = African American, HIS = Hispanic/Latino, EUR = European, ALL = All TOPMed MESA combined

isoforms (Apo E, Apo E2, Apo E3, Apo E4), C-Reactive Protein (CRP), Interleukin-1 receptor antagonist protein (Interleukin-1 receptor antagonist protein), Interleukin-6 receptor subunit alpha (IL-6 sRa), and Haptoglobin (Haptoglobin, Mixed Type). These are corroborated at the gene level by GWAS associations identified at the same locus. Eighteen of these protein-phenotype associations were significant SNP-phenotype associations in the original PAGE GWAS [1]. Matching our results, in other proteome studies using SOMAscan technology, isoforms of Apo E were associated with decreased HDL cholesterol, increased LDL cholesterol, and increased total cholesterol [30, 37].

In addition to the PAGE GWAS, independent GWAS have shown SNPs at the *APOE* locus associated with C-reactive protein [38–40], HDL cholesterol [38, 39, 41–44], LDL cholesterol [38, 39, 41–43, 45], and total cholesterol [38, 39, 41, 42, 46]. In our study, increased predicted abundance of CRP associated with increased measured C-reactive protein, effectively acting as a positive control for our method. Independent GWAS at the CRP locus show consistent associations with C-reactive protein measurement [38–40, 47–55]. Increased predicted IL-6 sRa associated with decreased C-reactive protein and the locus was previously implicated in other GWAS [38–40, 48, 49, 56].

Three of our protein-trait associations were not found in the original PAGE GWAS [1], but are still supported by independent GWAS. Increased Haptoglobin, Mixed Type was associated with decreased LDL cholesterol and decreased total cholesterol, both of which are

corroborated by GWAS at this locus [57]. Increased IL-1Ra was associated with decreased C-reactive protein. SNPs near IL-1Ra associated with C-reactive protein in an independent GWAS [49]. The directions of effect for each protein-phenotype association were consistent between all training populations.

## Most proteins remain predictable after adjusting for protein altering variants

All protein assays that rely on binding, including the SOMAscan assay used here, are susceptible to the possibility of binding-affinity effects, where protein-altering variants (PAVs) are associated with protein measurements due to differential binding rather than differences in protein abundance [26]. While we cannot differentiate these two possibilities, we can determine if SNP effects on protein abundance are independent of PAVs. We compared baseline elastic net models before and after adjusting protein abundance by any PAVs, which include frameshift variants, inframe deletions, inframe insertions, missense variants, splice acceptor variants, splice donor variants, splice region variants, start lost, stop gained, or stop lost.

We noted that the majority of results in Table 2 come from isoforms of Apo E, with replication among isoforms likely owing to known cross-reactivity of Apo E aptamers [26, 30, 37]. Abundance of each measured Apo E isoform associated with *APOE* genotype (Fig 6). Note that within each genotype, the target isoform abundances from the SOMAscan assay do not vary, indicating cross-reactivity effects are likely (Fig 6). Previous studies have found that protein levels of Apo E in plasma are correlated with the $\epsilon2$, $\epsilon3$, $\epsilon4$ haplotypes, but in the opposite direction than we observed [58–61]. After adjusting for the two missense SNPs (rs429358 and rs7412) that define these haplotypes, all protein-trait associations with Apo E fail to reach Bonferroni significance, indicating the well known $\epsilon2$, $\epsilon3$, $\epsilon4$ haplotypes drive the associations. Binding affinity differences among the haplotypes likely contribute, at least in part, to these protein-trait associations. Because *APOE* is a well known locus associated with many complex traits, these results demonstrates how SOMAscan-derived PWAS associations should be interpreted with caution (See Discussion).

Across all proteins, of the 1170 models built across all training populations, 39.8% of models remained unadjusted because they lacked a PAV in their 1 Mb cis-window (n = 466); 23.3% of models showed only marginal reduction in cross-validated $\rho$ after adjustment ($\Delta\rho <$ 0.1, n = 273); 12.6% of models showed a large decrease in model $\rho$, but retained significance ($\Delta\rho > 0.1$, n = 148); and 24.2% of models lost significance after adjustment and were not included in the final PAV-adjusted model sets (n = 283) (S9 Fig).

Among all five TOPMed MESA training populations, 701 protein predictions were made using baseline models in INTERVAL. Of these, 37.7% of models predicted in INTERVAL went unadjusted as they lacked a PAV (n = 264); 27.8% of models had a marginal decrease in performance ($\Delta\rho < 0.1$, n = 195); 7.0% of models had a larger decrease in performance, but maintained significance ($\Delta\rho > 0.1$, n = 49); and 27.5% of models lost significance and were not predicted in INTERVAL after adjusting for PAVs (n = 193; S10 Fig).

Before PAV adjustment, we found 21 distinct associations that met all three significance criteria of Bonferroni significance, colocalization, and replication in UKB+ (Table 2). All of the non-Apo E associations, including the CRP, IL-6 sRa, IL-1Ra associations with C-reactive protein and the Haptoglobin, Mixed Type associations with LDL and total cholesterol, remain significant after PAV adjustment. Thus, these protein-trait associations are not due to PAV binding-affinity effects (Table 2, S7 Table).

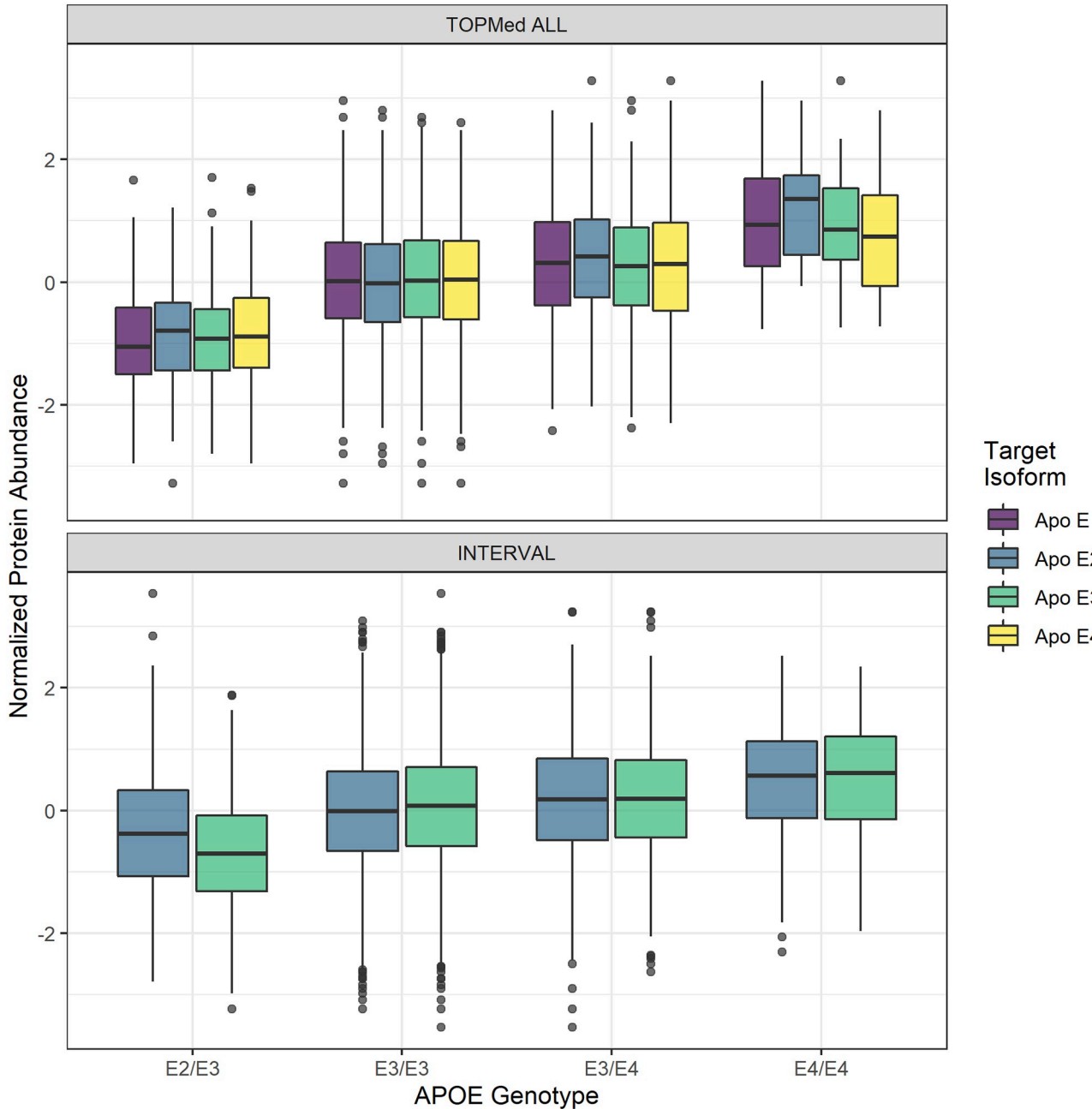

**Fig 6. Distribution of adjusted protein abundance.** We observe a linear association between *APOE* genotype and mean abundance of each Apo E isoform. Note that within a genotype, the target isoforms from the SOMAscan assay do not vary, indicating epitope cross-reactivity effects are likely. Top: Association in TOPMed ALL $\beta = 0.498$, $p = 4.60 \times 10^{-27}$. Bottom: Association in INTERVAL $\beta = 0.295$, $p = 1.98 \times 10^{-35}$. Only two isoforms were available in the INTERVAL dataset.

## Discussion

We built models for predicting protein abundances from genotypes in nearly 1000 African American, Chinese, European, and Hispanic/Latino individuals from TOPMed MESA for use in the PrediXcan framework. Protein abundances were measured on the SOMAscan platform using aptamer binding. We compared two strategies for constructing protein models,

preliminary fine-mapping followed by elastic net and baseline elastic net regression. Across all training populations and both model building strategies, 1187 unique protein aptamers have a significant prediction model ($\rho > 0.1$ and $p < 0.05$). We assessed model performance in the independent INTERVAL proteome population and in protein PrediXcan using GWAS summary statistics from the PAGE Study. Fine-mapping can improve cross-population prediction and maintains reliable replication of protein-trait pairs in PrediXcan compared to baseline elastic net proteome prediction. We found the most discoveries and reliable replications using ancestries-matched protein prediction models.

The ancestries of PAGE study participants most closely matched the ancestries of the TOPMed MESA AFA and HIS populations [1, 23]. We see increased discovery, colocalization, and replication when AFA and HIS protein models are used in S-PrediXcan compared to the larger EUR population protein models (Fig 5). Notably, all 3 populations, AFA, HIS, and EUR have similar numbers of significant protein models, especially after fine-mapping, even though the EUR population is 127% larger than AFA and 38% larger than HIS (Fig 1). Recent African ancestries populations like AFA and HIS have more SNPs and smaller LD blocks, which leads to both increased discovery and better fine mapping of the most likely causal SNPs [21, 22]. GWAS-based fine mapping from the PAGE Study demonstrated the value of leveraging diverse ancestries populations to improve causal SNP resolution prior to costly functional assays [1]. In our study, fine-mapping significantly improved the accuracy of cross-population prediction of protein abundance when training in AFA or CHN and testing in the European INTERVAL population (Fig 2). Models built in ALL performed better in INTERVAL than EUR-trained models for both fine-mapping and baseline strategies (Fig 3). However, fine-mapping in EUR did lead to more proteins that were predicted in INTERVAL than fine-mapping in ALL (340 vs. 259). Fine-mapping across ancestral populations likely leads to better performance when causal SNPs are shared among the populations. Thus, a combination of cross-ancestries and ancestries-matched fine-mapping will likely be necessary to optimize omics trait prediction in a locus-dependent manner.

Across all training populations, fine-mapped model building produced more models that passed our significance threshold of $\rho > 0.1$ and $p < 0.05$. We expected this result because we fine-mapped with all data and weighted SNPs by their PIPs prior to cross-validated elastic net modeling, i.e. 'double-dipping'. As our overall goal of building these models is the ability to test as many proteins as possible in PWAS, this double-dipping could be justified if it increased our ability to discover true associations, as was shown for TWAS [33]. Given that we tested more proteins with our fine-mapped model set, this technique did increase our ability to discover associations with S-PrediXcan compared to baseline (Fig 5A). However, when we assessed the reliability of these associations via colocalization and replication in independent studies, fine-mapped models and baseline models performed similarly (Fig 5B and 5C). While most fine-mapped PIPs were near zero in this study (S11 Fig), larger pQTL population sample sizes will result in more SNPs with larger PIPs, further homing in on causal SNPs in PWAS. Given the improved cross-population prediction of fine-mapped models (S7 Fig, S5 and S6 Tables) and their similar performance to baseline models in PWAS (Fig 5), we recommend using our fine-mapped models in PWAS. We also recommend population-matching in PWAS when protein model training sample sizes are within the same order of magnitude, as in TOPMed MESA, to maximize PWAS discovery, colocalization, and replication.

All protein assays that rely on binding are susceptible to the possibility of binding-affinity effects. A strong example of this issue is represented by Apo E, which has multiple isoforms measured in TOPMed MESA. SOMAscan aptamers that target isoforms of Apo E were previously shown to display cross-reactivity [26, 30, 37]. Thus, the aptamers do not distinguish

among the Apo E isoforms and instead might represent total Apo E abundance. But even if the isoform-derived aptamers are treated as total Apo E abundance measurements, inconsistencies with previous work arise.

In non-SOMAscan studies, the haplotype that determines the isoforms of Apo E was correlated with abundance of total Apo E in plasma, with $\epsilon2 > \epsilon3 > \epsilon4$ [58–61]. This is the opposite of what we observed here where individuals with the $\epsilon4$ allele have a greater measured abundance of Apo E than individuals with the $\epsilon2$ allele in both TOPMed MESA and INTERVAL (Fig 6). Other proteome studies using SOMAscan technology matched our results in that multiple aptamers of Apo E were associated with decreased HDL cholesterol, increased LDL cholesterol, and increased total cholesterol [30, 37]. However, *APOE* genotypes were not compared to protein abundance in the other SOMAscan studies [30, 37]. One possible explanation for our observed protein abundance vs. haplotye trend is that the E4 isoform has a greater binding affinity with all aptamers derived from Apo E proteins, possibly due to decreased glycosylation of the E4 isoform [59]. Additionally, the protein-trait associations we identified for Apo E proteins are driven by rs429358 and rs7412, indicating that differential abundance of these haplotypes is responsible for the associations found. It is not currently possible to differentiate between true differences in abundance of Apo E from differences in binding affinity among isoforms. The protein abundance mechanisms underlying the well established *APOE* genetic associations [1, 38–42, 46] remain to be elucidated.

Among other proteins, common (MAF >0.01) PAVs tend to be relatively rare. The majority of models we built either lack a PAV in their 1Mb cis-acting window or show only moderate changes in abundance due to PAVs. In addition, only 3.9% of proteins measured in TOPMed MESA share a genetic locus. This includes isoforms of the same protein as well as downstream products of the same precursor. A loss of association after PAV adjustment does not prove a false positive association due to PAV binding affinity effects. While possible, a loss of association after PAV adjument could also mean the PAVs are linked to a SNP functioning to affect protein abundance. However, if the association remains after PAV adjustment, we know binding affinity effects due to common PAVs are unlikely. Here, the CRP, IL-6 sRa, IL-1Ra associations with C-reactive protein and the Haptoglobin, Mixed Type associations with LDL and total cholesterol in PAGE and UKB+ remained significant after PAV adjustment. Thus, these protein-trait associations are not due to PAV binding-affinity effects. Follow up measurements of associated proteins with antibody-based assays would provide further independent validation of PWAS discoveries. While protein models can present unique challenges in interpretation, they are useful for discovery.

In addition to binding-affinity confounding, there are other limitations to our approach. The SOMAscan platform interrogates a subset of plasma proteins, and thus applying PrediXcan is not yet truly a proteome-wide association study. Protein measurement in other tissues is likely more appropriate than plasma for non-blood-related phenotypes. Proteins with low heritability or levels that fluctuate greatly in response to environmental stimuli are not well suited to the PWAS approach. Additionally, trans-acting SNPs were not included in this analysis, but may be useful for prediction, especially as proteome sample sizes increase. We demonstrated population-matched baseline protein prediction models map the most trait associations that replicate in larger populations. More genomes and proteomes in African ancestries and admixed populations are needed to improve fine-mapping protein model development and to better understand the mechanisms underlying complex traits in all populations.

## Materials and methods

### Ethics statement

This work was approved by the Loyola University Chicago Institutional Review Board (Project numbers 2014 and 2829). All data were previously collected and analyzed anonymously.

### Training data

**TOPMed MESA.** The Trans Omics for Precision Medicine (TOPMed) Consortium seeks to further elucidate the genetic architecture of several complex diseases including heart, lung, and sleep disorders through whole-genome sequencing, additional omics integration, and clinical phenotyping [62]. TOPMed includes data from a number of studies including MESA [31]. Samples from MESA were used to measure multiple omics traits in the TOPMed MESA Multi-omics Pilot Study [25]. Here, we used the TOPMed MESA proteomics data to train protein prediction models from genotypes. Protein levels were previously measured using a SOMAscan HTS Assay 1.3K for plasma proteins. The SOMAscan Assay is an aptamer based multiplex protein assay which measures protein levels by the number of protein specific aptamers which successfully bind to their target protein, though some proteins may be targeted by multiple aptamers [24, 25]. When more than one aptamer targets the same protein, each aptamer typically targets different isoforms of the same protein. In this study, each aptamer-based measurement is considered an independent protein. The TOPMed MESA training data we used includes genotypes and protein level measurements for four populations: African American (AFA, $n = 183$), Chinese (CHN, $n = 71$), European (EUR, $n = 416$), and Hispanic/Latino (HIS, $n = 301$). In addition to these we also consider a multi-ethnic population comprised of all four populations combined (ALL, $n = 971$).

### Test data

**INTERVAL.** Our test data come from the INTERVAL study, comprised of 3,301 individuals of European ancestries with both genotype (EGAD00010001544) and blood plasma aptamers levels as measured by a SOMAscan assay (EGAD00001004080) [26, 34, 63]. The SOMAscan assay employed by INTERVAL measured 3,622 proteins measured [63]. Data generation and quality control have been previously described in detail [26, 34]. Genotyping was performed using an Affymetrix Axiom UK Biobank genotyping array and imputed on the Sanger imputation server using a combined 1000 Genomes Phase 3-UK10K reference panel [26, 64]. We used genotypes with MAF > 0.01, $R^2 > 0.8$. Protein abundances were previously log transformed, adjusted for age, sex, duration between blood draw and processing (binary, $\leq$ 1 day/ >1 day) and the first three genetic principal components [26]. We used the rank normalized residuals from this linear regression as our measure of protein abundance.

### TOPMed genotype QC

Genotypes and measured protein aptamer levels were available for 971 individuals. Genotype data were accessed via the MESA SHARe study (phs000420.v6.p3) and were imputed on the Michigan imputation server (Minimac4.v1.0.0) using the 1000 Genomes reference panel [15]. We calculated $F_{ST}$ between each TOPMed population and INTERVAL using PLINK [65, 66]. Genotypes in each individual population were filtered for imputation $R^2 > 0.8$, $MAF > 0.01$. The multiethnic ALL population genotypes were filtered to the intersection of SNPs with imputation $R^2 > 0.8$ in all four individual populations and $MAF > 0.01$ across all 971 individuals. We used the genotype dosages as predictors in our regression analyses [65–67].

We used PCAIR as implemented in the GENESIS library in R to calculate robust estimates of principal components in the presence of cryptic relatedness [68, 69]. Prior to calculating principal components, the KING algorithm makes robust estimates of the pairwise kinship matrix within a population [70, 71]. Then, the PCAIR algorithm partitions data into a set of mutually unrelated individuals used to estimate principal components and a set of related individuals whose eigenvectors are imputed on the basis of kinship measures. We calculated principal components within each population and in the ALL population for use in protein prediction model building. The partition of related individuals contained 1 person within AFA, 2 people within CHN, 5 in EUR, and 25 in HIS. Within the ALL population 44 people were contained within the related partition. We also calculated principal components including ALL and 1000 Genomes reference populations to visualize population structure across MESA (S1 Fig).

## TOPMed protein aptamer level QC

Protein levels were measured at two time points, Exam 1 and Exam 5 of MESA. Similar to a previous SOMAscan protein study [26], we log transformed each time point and then adjusted for age and sex. We then took the mean of the two time points (if a participant was not measured at both time points then we treated the measured time point as their mean), performed rank inverse normalization, and adjusted for the first ten genotypic principal components prior to downstream modeling.

## pQTL fine mapping

We used Matrix eQTL [72] to perform a genome wide cis-acting pQTL analysis in each population (AFA, CHN, EUR, and HIS) as well as in all four populations combined (ALL). We performed association testing using the protein aptamer level adjusted for age, sex, and 10 genotypic principal components as the response and SNPs as the predictors. We defined the cis-acting SNPs as those within 1 Mb of the TSS of the gene corresponding to the aptamer. Aptamers may map to more than one gene as in the case the aptamer binds to a protein complex. However, for all analyses done here, we treated these multiple cis-windows as independent loci and estimate these cis-effects separately for each gene to which an aptamer maps. For those aptamers which map to multiple genes, each aptamer-gene pair is treated as an independent phenotype with identical values.

We performed fine mapping using the software tool DAP-G [73, 74]. After identifying cis-pQTLs, prior probabilities are estimated from pQTL data using the software tool torus [75]. These priors are then used by the DAP-G algorithm to estimate the PIP of a given SNP within a particular cis-window as likely causal (or tightly linked to the causal SNP) for the protein in question. We note that without a functional assay, a causal SNP cannot be distinguished from a proxy SNP. As in pQTL discovery, fine mapping is done independently for each gene to which an aptamer maps. Aptamer level annotations were created by mapping proteins to genomic coordinates using GENCODE (GRCh38), version 32 (Ensembl 98) [76].

## Elastic net regression

In all five training populations (AFA, ALL, CHN, EUR, and HIS) we performed nested cross-validated elastic net regression [77] with mixing parameter $\alpha = 0.5$ using genotype dosages within the 1 Mb cis-window as predictors and the adjusted protein aptamer levels as response. Models were trained using the *glmnet* package in R [78]. We used nested cross-validation to calculate cross validated Spearman correlation ($\rho$) between predicted and observed protein levels as our metric of model performance using 5 folds in our outer loop with the λ that

minimizes the cross validated error estimated by 10-fold cross validation in our inner loop. The final model for testing in INTERVAL and use in PWAS is then fit on all data with lambda chosen by 10 fold cross validation. As a measure of model quality, using the same thresholds used in PrediXcan transcriptome modeling [11, 33], we filtered each model set to include those protein models with a cross-validated $\rho > 0.1$ and $p < 0.05$. We term models built in this manner as "baseline" elastic net models.

In addition to the baseline elastic net models, we trained elastic models using the fine-mapped PIPs as penalty factors as described in Barbeira et al. 2020 [33]. A penalty factor of 0 for a particular SNP will result in that SNP always being kept in the model while a higher penalty factor will result in that SNP being less likely to be included in the model. We use $1-PIP$ as penalty factors for elastic net regression. The higher the PIP, the more likely the SNP associates with protein and the lower the penalty factor, or the more likely that SNP is kept in the regression model. We test three thresholds of minimum PIP for each SNP to be considered as a predictor for a protein: $PIP > 0$, $PIP > 0.001$, and $PIP > 0.01$. In each case, we only included those SNPs with a PIP higher than the given threshold as predictors for a given protein. Additionally, DAP-G assigns SNPs to clusters based on LD. We employ two strategies for handling these clusters. First, as SNPs within a cluster are correlated, we filter these clusters to only include the SNP with the highest PIP. These SNPs which pass our PIP threshold are then used for elastic net regression. Second, we do no filtering based on cluster and use all SNPs that pass the PIP threshold are then used for elastic net regression. See S1 Table for a summary of all the model sets built as well as notation.

## Heritability estimation

We used the software GEMMA [79] to implement BSLMM [35] for each protein aptamer with 100K sampling steps per aptamer. BSLMM estimates the PVE (the proportion of variance in phenotype explained by the additive genetic model, analogous to $h^2$). From the second half of the sampling iterations for each aptamer, we compared the median and the 95% credible sets of the PVE to model performance in INTERVAL.

## Protein altering variants

Protein assays that rely on binding are susceptible to the possibility of binding-affinity effects. SNPs in a protein's aptamer binding site may affect subsequent protein level measurement. Following the convention of Sun et al., we term Protein Altering Variants (PAVs) as SNPs which may result in differential binding to the target aptamer [26]. We use the the Ensembl VEP v100.2 tool to annotate variants using the "per gene" option [80, 81]. PAVs are variants annotated as one of the following: consequence in coding sequence variant, frameshift variant, inframe deletion, inframe insertion, missense variant, protein altering variant, splice acceptor variant, splice donor variant, splice region variant, start lost, stop gained, or stop lost. To address the possibility of binding affinity effects we built additional models that adjust for PAVs. For each protein, we extracted the matrix of PAV genotypes and used this to perform principal component analysis. We use the number of PCs which account for 95% of variance in the matrix of PAV genotypes to adjust the protein abundance. We used the residuals of this linear regression as the adjusted protein abundance. We removed the PAVs from the genotype matrix and then performed elastic net regression on the adjusted protein abundance. If no PAVs that pass genotype QC were in the 1Mb cis-window, we made no adjustment and reran the baseline elastic net regression. We compared adjusted models to unadjusted models to determine if the prediction was driven by the PAVs (reduced correlation) or SNPs independent of the PAVs (similar correlation). Reduced correlation in the adjusted model could be

due to binding affinity effects or could mean the PAVs are linked to a SNP functioning to affect protein abundance.

## Adjustment for Apo E haplotypes

The PAVs which define isoforms of Apo E (rs429358 and rs7412) are well known loci which associate with Alzheimer's Disease and cholesterol phenotypes [1, 38, 39, 41–44, 82–84]. The $\epsilon 2$ allele is defined by the T-T haplotype, $\epsilon 3$ by T-C, and $\epsilon 4$ by C-C at rs429358 and rs7412, respectively. Because rs429358 and rs7412 did not pass genotype QC in all training populations due to imputation $R^2 < 0.8$, they were not included in our elastic net modeling and fine-mapping. However, both SNPs had imputation $R^2 > 0.4$ in all populations, so we used the imputed genotypes to examine the effect of of PAV adjustment at this important locus.

## Out of sample testing in INTERVAL

We obtained measurements of protein abundance that were previously natural log-transformed; adjusted for age, sex, duration between blood draw and processing, and the first 3 genetic principal components; and rank-inverse normalized [26]. We predicted protein abundance in the INTERVAL cohort using models built in each TOPMed MESA population. We used the Spearman correlation between the predicted abundance for a protein and the observed abundance for a protein as our measure of prediction accuracy. Of the proteins measured in INTERVAL, 804 protein aptamers mapped uniquely to an aptamer measured in TOPMed.

## Proteome-wide association studies

To study the utility of our protein predictive models for association studies, we ran S-PrediXcan using GWAS summary statistics derived from the Population Architecture using Genomics and Epidemiology (PAGE) study [1, 32, 36]. PAGE is a large cohort of multi-ethnic, non-European ancestries comprising 49,839 individuals with summary statistics available from the GWAS Catalog for 28 clinical and behavioral phenotypes. Individuals in PAGE self-identified as African American/Afro-Caribbean, Hispanic/Latin American, Oceanian, Hawaiian, and Native American [1, 36]. We performed S-PrediXcan the find protein associations with the PAGE 28 phenotypes using protein prediction models from each TOPMed MESA population. We considered protein-trait associations significant if they met the Bonferroni significance threshold calculated by counting all association tests performed for a given model, i.e., baseline or fine-mapped. For example, for the baseline model sets, all association tests for all populations and all phenotypes were pooled, and the Bonferroni threshold was calculated as 0.05/$n_{tests}$. This threshold was calculated independently for each model building strategy ($p < 1.54 \times 10^{-6}$ for baseline, $p < 7.60 \times 10^{-7}$ for fine-mapped).

## Colocalization

We applied the software COLOC [32, 85–87] to our TOPMed pQTL summary statistics and PAGE GWAS summary statistics [1] to determine if pQTLs and GWAS hits are colocalized. We used COLOC version 4.0–4 [87], which allows user inputted LD correlation matrices for interpreting LD patterns at certain loci. Using SNPs within 1Mb of the transcription start and end sites of each protein-coding gene, we built LD correlation matrices from TOPMed MESA for our COLOC analyses using PLINK [65, 66]. COLOC outputs posterior probabilities (P) for each of their five hypotheses. A high P4 probability ($P4 > 0.5$) suggests that the pQTL and GWAS signals are colocalized while a P3 probability greater than 0.5 indicates likely

independent pQTL and GWAS signals. P0, P1, and P2 values greater than 0.5 indicate an unknown association [32, 87]. COLOC version 4.0–4 allows users to relax the assumption that there is only a single independent association for each phenotype tested and outputs SNP-level results for multiple variants. For this analysis, each protein-level needs only one set of variants to have $P4 > 0.5$ for it to be considered significantly colocalized with a phenotype. We determined if a protein-level has colocalized or independent signals by looking at the highest P4 value.

## Replication

To test protein-trait associations discovered in PAGE for replication, we performed S-PrediXcan with GWAS summary statistics from the UKBiobank with the same or similar phenotypes as those included in PAGE [1, 2]. However, some PAGE phenotypes were not tested in the available UKBiobank GWAS (http://www.nealelab.is/uk-biobank/) [2], thus we performed S-PrediXcan in an available GWAS with a large European sample size for the same or similar trait as the PAGE phenotype (S2 Table) [3–10]. For this reason, we refer to this set of GWAS as UKB+.

We examine only our colocalized, S-PrediXcan significant associations in PAGE for replication in UKB+. We define an association as replicated if the same association is also S-PrediXcan Bonferroni significant ($p < 1.54 \times 10^{-6}$ for baseline, $p < 9.59 \times 10^{-7}$ for fine-mapped) in UKB+ and has the same direction of effect.

## Supporting information

**S1 Fig. Genotype principal component analysis.** Biplot of the first two principal components of TOPMed MESA populations with 1000 Genomes reference populations. Genetic PCs of TOPMed participants with both genomic and proteomic data were estimated with PCAIR. Pop codes: TOPMed African American (AFA), TOPMed Chinese (CHN), TOPMed European (EUR), TOPMed Hispanic (HIS), 1000 Genomes East Asians from Beijing, China and Tokyo, Japan (ASN), 1000G European ancestry from Utah (CEU), and 1000G Yoruba from Ibadan, Nigeria (YRI).
(TIF)

**S2 Fig. pQTLs are enriched near the TSS.** Significant pQTL (FDR <0.05) effect sizes are plotted versus the SNP distance to the TSS of the protein encoding gene in each TOPMed MESA population. Contour lines from two-dimensional kernel density estimation show pSNPs are concentrated at the TSS in all populations.
(TIF)

**S3 Fig. Protein prediction model counts.** In total 1238 unique protein aptamers have significant prediction models ($\rho > 0.1$, $p < 0.05$) across all strategies and training populations. Number of significant protein models scales approximately with sample size of the training population, with the exception of ALL fine-mapped models.
(TIF)

**S4 Fig. Protein prediction model performance correlations.** The pairwise Pearson correlations between prediction performance of each model building strategy trained in each TOPMed MESA population. Prediction performance is the Spearman correlation between observed and predicted expression in the independent INTERVAL study. Note, most fine-mapped models within a population had high correlation, with slightly reduced correlations

between fine-mapped (LD cluster filtered true) and baseline models. See S1 Table for model notations.
(TIF)

**S5 Fig. Protein prediction model performance correlates with protein heritability.** Comparison of the BSLMM PVE (pve50) by cis-SNPs for each protein trait in each population to the prediction performance in INTERVAL ($\rho$). Gray vertical lines represent the 95% credible set for each PVE estimate and the blue line is the linear regression fit.
(TIF)

**S6 Fig. Fine-mapped to baseline model comparisons.** Vertical axis is the fine mapped model performance when predicting in INTERVAL. Horizontal axis is the baseline elastic net model performance when predicting in INTERVAL. Each dot represents a protein that is predicted by both baseline models and fine mapped models. Performance is measured as the Spearman correlation between the measured protein aptamer level and the predicted protein aptamer level.
(TIF)

**S7 Fig. Population specific performance in an independent cohort.** We compare the performance of our different training populations at predicting in INTERVAL, a predominantly European cohort. For a particular model building strategy we first take the intersection of proteins that are predicted by all five training populations and then test for differences in the distribution of Spearman correlations by ANOVA and permuted F-test. We find a significant difference among training populations for our baseline elastic net models (30 proteins, F = 13.30, p = 5.93e-09), 0.001_F models (61 proteins, F = 3.41, p = 0.0098), and 0_F models (59 proteins, F = 3.54, p = 0.0080).
(TIF)

**S8 Fig. Significant PWAS association counts in PAGE.** Fine-mapped model sets consistently have a greater number of Bonferroni significant associations than baseline model sets. However when including significant evidence of colocalization by COLOC and replication status as additional significance criteria, baseline has a higher number of significant associations.
(TIF)

**S9 Fig. Comparison of protein altering variant (PAV) adjusted baseline models to unadjusted baseline models.** Cross-validated rho within each TOPMed MESA population is plotted on both axes. PAV adjusted model sets are on the Y axis, while standard model sets are plotted on the X axis. Most models were unadjusted for PAVs as the protein does not contain a PAV (yellow points).
(TIF)

**S10 Fig. Performance of PAV adjusted model sets vs unadjusted model sets in INTERVAL.** Prediction performance rho in INTERVAL using models built in each TOPMed MESA population is plotted. PAV adjusted model sets are on the Y axis, while standard model sets are plotted on the X axis. Most models were unadjusted for PAVs as the protein does not contain a PAV (yellow points). Most models are either unadjusted (yellow) or have only a small decrease in performance. 7.0% of models had a larger decrease in performance (change in $\rho > 0.1$), but maintained significance. Not plotted here is the 23.6% of models which are significant in our unadjusted regression, but are no longer significant in our PAV adjusted regression.
(TIF)

**S11 Fig. Distribution of protein-associated SNP posterior inclusion probabilities (PIPs).**
The vast majority of PIPs used to calculate penalty factors in our fine-mapped models are near
0. A) Distribution of PIPs >0 B) PIPs >0.001 C) PIPs >0.01.
(TIF)

**S1 Table. Protein prediction model notation.** For each training population, we built seven
types of model for comparison. One standard elastic net regression, and six fine-mapped
model sets with variable PIP threshold and LD filtering strategies. For fine-mapped models,
SNPs must meet the minimum PIP threshold specified to be included as predictors. Addition-
ally as our fine mapping software, DAP-G, clusters SNPs according to LD, we optionally filter
clusters to only include the SNP with the highest PIP.
(XLSX)

**S2 Table. UKB+ data.** Sources for GWAS summary statistics comprising our UKB+ data.
Where possible we use GWAS summary statistics generated using the UKB. However, when a
phenotype is not available, we sourced data from the GWAS catalogue for other large Euro-
pean GWAS.
(XLSX)

**S3 Table. Proteins not in MASHR summaries.** Model summaries for all proteins that do not
have an RNA equivalent model for either Whole Blood models or any tissue as published in
Barbeira et al 2020 GTEx v8 MASHR models. In total 19 distinct protein aptamers do not have
an RNA equivalent model across any tissue model from Barbeira et al. 2020 GTEx v8 MASHR
models. 424 aptamers do not have an RNA equivalent model in Whole Blood models from
Barbeira et al. 2020 GTEx v8 MASHR models.
(XLSX)

**S4 Table. Fine-mapped to baseline paired t-test statistics.** Test statistics and p values for
model comparisons between fine-mapping strategies and baseline elastic net models. Fine-
mapped models in AFA consistently outperformed baseline models. Fine-mapped CHN was
either significantly better or not different. Fine-mapped ALL, HIS, and EUR were either signif-
icantly worse or not different.
(XLSX)

**S5 Table. Population specific performance comparison statistics.** Test statistics for ANOVA
and permuted F test comparing the predictive performance of different training populations
for a particular model building strategy. ANOVA is run using the training population and the
aptamer model ID as factors and Spearman Correlation as response. For our permuted F test
the aptamer model ID is treated as a blocking factor for permutation.
(XLSX)

**S6 Table. Tukey's HSD for population differences.** Results of Tukey's HSD for model
building strategies that showed a significant difference in training populations by ANOVA.
For baseline elastic net models, EUR, HIS, and ALL were all significantly greater than CHN
and AFA with all other pairs not significantly different. For 0.001_F models only HIS
was greater than CHN with all other pairs not significantly different. For 0_F models both
HIS and ALL were significantly greater than CHN with all other pairs not significantly dif-
ferent.
(XLSX)

**S7 Table. List of colocalized, S-PrediXcan significant associations in PAGE.** Across all
model building strategies and training populations we identify 27 distinct associations that are

both S-PrediXan significant and with significant evidence of colocalization. This spans 11 unique protein models and 8 phenotypes.
(XLSX)

**S8 Table. List of NHLBI TOPMed consortium members.**
(XLSX)

## Acknowledgments

We gratefully acknowledge all participants in TOPMed MESA, INTERVAL, PAGE, and the 1000 Genomes Project. We also thank all members of the NHBLI TOPMed Consortium (S8 Table).

## Author Contributions

**Conceptualization:** Ryan Schubert, Heather E. Wheeler.

**Data curation:** Robert Gerszten, W. Craig Johnson, Xiuqing Guo, Matthew Conomos.

**Formal analysis:** Ryan Schubert, Elyse Geoffroy, Isabelle Gregga, Ashley J. Mulford, Heather E. Wheeler.

**Funding acquisition:** Stephen S. Rich, Jerome I. Rotter, Heather E. Wheeler.

**Methodology:** Ryan Schubert, Stephen S. Rich, Ani Manichaikul, Hae Kyung Im.

**Project administration:** Jerome I. Rotter, Heather E. Wheeler.

**Resources:** Francois Aguet, Kristin Ardlie, Robert Gerszten, Clary Clish, David Van Den Berg, Kent D. Taylor, Peter Durda, Elaine Cornell, Xiuqing Guo, Yongmei Liu, Russell Tracy, Tom Blackwell, George Papanicolaou, Tuuli Lappalainen, Anna V. Mikhaylova, Timothy A. Thornton, Michael H. Cho, Christopher R. Gignoux, Leslie Lange, Ethan Lange, Stephen S. Rich, Jerome I. Rotter, Ani Manichaikul, Hae Kyung Im, Heather E. Wheeler.

**Visualization:** Ryan Schubert, Heather E. Wheeler.

**Writing – original draft:** Ryan Schubert, Heather E. Wheeler.

**Writing – review & editing:** Ryan Schubert, Elyse Geoffroy, Isabelle Gregga, Ashley J. Mulford, Tom Blackwell, Tuuli Lappalainen, Anna V. Mikhaylova, Timothy A. Thornton, Michael H. Cho, Christopher R. Gignoux, Leslie Lange, Ethan Lange, Stephen S. Rich, Jerome I. Rotter, Ani Manichaikul, Hae Kyung Im, Heather E. Wheeler.

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
