## [Decision Letter · Decision Letter 0]

19 Oct 2021

PONE-D-21-26641Protein prediction for trait mapping in diverse populationsPLOS ONE

Dear Dr. Wheeler,

Thank you for submitting your manuscript to PLOS ONE. After careful consideration, we feel that it has merit but does not fully meet PLOS ONE’s publication criteria as it currently stands. Therefore, we invite you to submit a revised version of the manuscript that addresses the points raised during the review process.

We look forward to receiving your revised manuscript.

Kind regards,

Heming Wang, PhD

Academic Editor

PLOS ONE

Journal Requirements:

"This work is supported by the NIH National Human Genome Research Institute 

Academic Research Enhancement Award R15 HG009569 (HEW). 

MESA and the MESA SHARe project are conducted and supported by the National 

Heart, Lung, and Blood Institute (NHLBI) in collaboration with MESA investigators. 

Support for MESA is provided by contracts 75N92020D00001, HHSN268201500003I, 

N01-HC-95159, 75N92020D00005, N01-HC-95160, 75N92020D00002, N01-HC-95161,

75N92020D00003, N01-HC-95162, 75N92020D00006, N01-HC-95163, 75N92020D00004, 

N01-HC-95164, 75N92020D00007, N01-HC-95165, N01-HC-95166, N01-HC-95167, 

N01-HC-95168, N01-HC-95169, UL1-TR-000040, UL1-TR-001079, UL1-TR-001420. 

Funding for SHARe genotyping was provided by NHLBI Contract N02-HL-64278. 

Genotyping was performed at Affymetrix (Santa Clara, California, USA) and the Broad 

Institute of Harvard and MIT (Boston, Massachusetts, USA) using the Affymetrix 

Genome-Wide Human SNP Array 6.0. MESA Family is conducted and supported by 

the National Heart, Lung, and Blood Institute (NHLBI) in collaboration with MESA 

investigators. Support is provided by grants and contracts R01HL071051, 

R01HL071205, R01HL071250, R01HL071251, R01HL071258, R01HL071259, by the 

National Center for Research Resources, Grant UL1RR033176. Also supported in part

by the National Center for Advancing Translational Sciences, CTSI grant 

UL1TR001881, and the National Institute of Diabetes and Digestive and Kidney 

Disease Diabetes Research Center (DRC) grant DK063491 to the Southern California 

Diabetes Endocrinology Research Center. 

The TOPMed MESA Multi-Omics project was conducted by the University of 

Washington and LABioMed (HHSN2682015000031/HHSN26800004). Molecular data for 

the Trans-Omics in Precision Medicine (TOPMed) program was supported by the

National Heart, Lung and Blood Institute (NHLBI). SOMAscan proteomics for NHLBI 

TOPMed: Multi-Ethnic Study of Atherosclerosis (MESA) (phs001416.v1.p1) was 

performed at the Broad Institute and Beth Israel Proteomics Platform 

(HHSN268201600034I). Core support including centralized genomic read mapping and 

genotype calling, along with variant quality metrics and filtering were provided by the 

TOPMed Informatics Research Center (3R01HL-117626-02S1; contract 

HHSN268201800002I). Core support including phenotype harmonization, data 

management, sample-identity QC, and general program coordination were provided by

the TOPMed Data Coordinating Center (R01HL-120393; U01HL-120393; contract 

HHSN268201800001I). We gratefully acknowledge the studies and participants who 

provided biological samples and data for TOPMed. 

Participants in the INTERVAL randomised controlled trial were recruited with the 

active collaboration of NHS Blood and Transplant England (www.nhsbt.nhs.uk), which 

has supported field work and other elements of the trial. DNA extraction and 

genotyping was co-funded by the National Institute for Health Research (NIHR), the 

NIHR BioResource (http://bioresource.nihr.ac.uk) and the NIHR [Cambridge August 17, 2021 15/25

Biomedical Research Centre at the Cambridge University Hospitals NHS Foundation 

Trust]. The INTERVAL study was funded by NHSBT (11-01-GEN). The academic 

coordinating centre for INTERVAL was supported by core funding from: NIHR Blood 

and Transplant Research Unit in Donor Health and Genomics (NIHR 

BTRU-2014-10024), UK Medical Research Council (MR/L003120/1), British Heart 

Foundation (SP/09/002; RG/13/13/30194; RG/18/13/33946) and the NIHR 

[Cambridge Biomedical Research Centre at the Cambridge University Hospitals NHS 

Foundation Trust]. Proteomic assays were funded by the academic coordinating centre 

for INTERVAL and MRL, Merck & Co., Inc. A complete list of the investigators and 

contributors to the INTERVAL trial is provided in Di Angelantonio et al. [34]. The 

academic coordinating centre would like to thank blood donor centre staff and blood 

donors for participating in the INTERVAL trial. 

This work was supported by Health Data Research UK, which is funded by the UK 

Medical Research Council, Engineering and Physical Sciences Research Council, 

Economic and Social Research Council, Department of Health and Social Care 

(England), Chief Scientist Office of the Scottish Government Health and Social Care 

Directorates, Health and Social Care Research and Development Division (Welsh 

Government), Public Health Agency (Northern Ireland), British Heart Foundation and 

Wellcome. 612

The views expressed are those of the authors and not necessarily those of the NHS, 

the NIHR or the Department of Health and Social Care. The funders had no role in 

study design, data collection and analysis, decision to publish, or preparation of the 

manuscript."

"This work is supported by the NIH National Human Genome Research Institute Academic Research Enhancement Award R15 HG009569 (HEW).

MESA and the MESA SHARe project are conducted and supported by the National Heart, Lung, and Blood Institute (NHLBI) in collaboration with MESA investigators. Support for MESA is provided by contracts 75N92020D00001, HHSN268201500003I, N01-HC-95159, 75N92020D00005, N01-HC-95160, 75N92020D00002, N01-HC-95161, 75N92020D00003, N01-HC-95162, 75N92020D00006, N01-HC-95163, 75N92020D00004, N01-HC-95164, 75N92020D00007, N01-HC-95165, N01-HC-95166, N01-HC-95167, N01-HC-95168, N01-HC-95169, UL1-TR-000040, UL1-TR-001079, UL1-TR-001420. Funding for SHARe genotyping was provided by NHLBI Contract N02-HL-64278. Genotyping was performed at Affymetrix (Santa Clara, California, USA) and the Broad Institute of Harvard and MIT (Boston, Massachusetts, USA) using the Affymetrix Genome-Wide Human SNP Array 6.0. MESA Family is conducted and supported by the National Heart, Lung, and Blood Institute (NHLBI) in collaboration with MESA investigators. Support is provided by grants and contracts R01HL071051, R01HL071205, R01HL071250, R01HL071251, R01HL071258, R01HL071259, by the National Center for Research Resources, Grant UL1RR033176. Also supported in part by the National Center for Advancing Translational Sciences, CTSI grant UL1TR001881, and the National Institute of Diabetes and Digestive and Kidney Disease Diabetes Research Center (DRC) grant DK063491 to the Southern California Diabetes Endocrinology Research Center.

The TOPMed MESA Multi-Omics project was conducted by the University of Washington and LABioMed (HHSN2682015000031/HHSN26800004). Molecular data for the Trans-Omics in Precision Medicine (TOPMed) program was supported by the National Heart, Lung and Blood Institute (NHLBI). SOMAscan proteomics for NHLBI TOPMed: Multi-Ethnic Study of Atherosclerosis (MESA) (phs001416.v1.p1) was performed at the Broad Institute and Beth Israel Proteomics Platform (HHSN268201600034I). Core support including centralized genomic read mapping and genotype calling, along with variant quality metrics and filtering were provided by the TOPMed Informatics Research Center (3R01HL-117626-02S1; contract HHSN268201800002I). Core support including phenotype harmonization, data management, sample-identity QC, and general program coordination were provided by the TOPMed Data Coordinating Center (R01HL-120393; U01HL-120393; contract HHSN268201800001I). We gratefully acknowledge the studies and participants who provided biological samples and data for TOPMed.

Participants in the INTERVAL randomised controlled trial were recruited with the active collaboration of NHS Blood and Transplant England (www.nhsbt.nhs.uk), which has supported field work and other elements of the trial. DNA extraction and genotyping was co-funded by the National Institute for Health Research (NIHR), the NIHR BioResource (http://bioresource.nihr.ac.uk) and the NIHR [Cambridge Biomedical Research Centre at the Cambridge University Hospitals NHS Foundation Trust]. The INTERVAL study was funded by NHSBT (11-01-GEN). The academic coordinating centre for INTERVAL was supported by core funding from: NIHR Blood and Transplant Research Unit in Donor Health and Genomics (NIHR BTRU-2014-10024), UK Medical Research Council (MR/L003120/1), British Heart Foundation (SP/09/002; RG/13/13/30194; RG/18/13/33946) and the NIHR [Cambridge Biomedical Research Centre at the Cambridge University Hospitals NHS Foundation Trust]. Proteomic assays were funded by the academic coordinating centre for INTERVAL and MRL, Merck & Co., Inc. A complete list of the investigators and contributors to the INTERVAL trial is provided in Di Angelantonio et al. The academic coordinating centre would like to thank blood donor centre staff and blood donors for participating in the INTERVAL trial.

This work was supported by Health Data Research UK, which is funded by the UK Medical Research Council, Engineering and Physical Sciences Research Council, Economic and Social Research Council, Department of Health and Social Care (England), Chief Scientist Office of the Scottish Government Health and Social Care Directorates, Health and Social Care Research and Development Division (Welsh Government), Public Health Agency (Northern Ireland), British Heart Foundation and Wellcome.

The views expressed are those of the authors and not necessarily those of the NHS, the NIHR or the Department of Health and Social Care. The funders had no role in study design, data collection and analysis, decision to publish, or preparation of the manuscript"

Reviewers' comments:

Reviewer's Responses to Questions

**Comments to the Author**

1. Is the manuscript technically sound, and do the data support the conclusions?

Reviewer #1: Yes

Reviewer #2: Yes

2. Has the statistical analysis been performed appropriately and rigorously? 

Reviewer #1: Yes

Reviewer #2: Yes

3. Have the authors made all data underlying the findings in their manuscript fully available?

Reviewer #1: Yes

Reviewer #2: Yes

4. Is the manuscript presented in an intelligible fashion and written in standard English?

Reviewer #1: Yes

Reviewer #2: Yes

5. Review Comments to the Author

Reviewer #1: Review: Protein prediction for trait mapping in diverse populations

Overview: Schubert et al present work on predicting protein abundance in the TOPMed data using genotype information into large-scale GWAS across diverse populations. Their work builds conceptually on the TWAS model, which predicts steady state gene expression (mRNA levels) rather than downstream protein abundance. They investigated the predictive performance and transferability of prediction models across four TOPMed MESA populations (African Americans, Chinese, European, and Hispanic/Latino). Particular attention was paid to the performance of fine-mapping or baseline prediction models across populations, and follow-up with out-of-sample performance in the INTERVAL protein study. Next, the applied their prediction models to GWAS data for 28 phenotypes of diverse populations in the PAGE consortium.

I find this area to be interesting and useful, however I find the results as presented to provide very little insight and lack clear take-aways for downstream decision making. For example, the authors performed a good deal of analyses to quantify protein prediction accuracy, but fail to provide a clear recommendation on which approach to use (e.g., does having more models matter upfront, or having population matched model/GWAS downstream). I appreciate the complexity of multiple analyses across diverse populations complicates matters, but simple meta-analysis tools can simplify the big picture and present results in a consistent manner that help the reader understand what approaches work better on average. Similarly, there are a number of analyses that should have been performed to help place findings in context prior to predictive modeling. Overall, I think the data generated in this manuscript to be interesting and valuable to the broader community, but that a simplified presentation could greatly help with communicating primary findings and informing downstream TWAS/PWAS users.

I provide more details below.

Major Comments:

1. Given that protein prediction models rely on pQTL signals, it would greatly help if the authors also discussed raw pQTL association (and fine-mapping) results across populations before discussing prediction. Understanding functional enrichment of protein regulatory mechanisms is interesting and crucially unexplored in data at this scale. Including these analyses would help provide context for downstream prediction models and shed light on regulatory mechanisms themselves, prior to their relationship with complex disease risk.

2. Similarly, prediction accuracy is inherently tied to heritability. It would be helpful to see how prediction accuracy tracks with in-sample h2g estimates (or out-of-sample INTERVAL R2 with INTERVAL h2g).

3. Can the authors provide some supporting analyses for when genes fail to replicate across populations? It would be interesting to see how avg Fst at a gene/locus tracks with avg cross-pop R2.

Reviewer #2: The authors took up an exploratory study that constructed pQTL models using TOPMed MESA cohorts of various ancestries, including African American, Chinese, European, Hispanic/Latino, and cross-population; models were further evaluated and validated using an independent cohort European INTERVAL. For each population-specific cohort or cross-population cohort, the authors have also developed a baseline model (which I believe was inclusive of all sequenced or genotyped variants) and a fine-mapped model. In general, fine-mapped models outperformed baseline models in terms of significant pQTL signals/models. Furthermore, the authors used the constructed models to perform PWAS on the PAGE cohort and replicated in the UKB+ data when the testing trait is available in the replication cohort. The authors successfully identified several known associations, for example, HDL-APOE.

1. In line 86-90, the authors stated that they identified 372 protein aptamers distinct to MESA and not found in GTEx Whole Blood models. Can there be false positives? Are these protein aptamers population-specific or from the cross-population model? It would explain the distinctiveness of these aptamers if these were population specific.

2. For table 1, are these all replication of previous findings? Are some of them novel? There was analysis in the result saying that some significant association signals of APOE isoforms went away after adjusted for PAV. Would it be better if this is noted in the table 1 or at least stated in the table 1 legend?

3. Were there any related samples in MESA? Did the authors adjust for relatedness among samples?

Minor suggestions or side questions

1. What the relationships between pQTLs in this study and MESA eQTLs? Is it correlated?

2. Some acronyms, like PIP, were declared more than once.

6. PLOS authors have the option to publish the peer review history of their article (what does this mean?). If published, this will include your full peer review and any attached files.

Reviewer #1: No

Reviewer #2: **Yes: **Binglan Li

---

## [Author Response · Author response to Decision Letter 0]

16 Dec 2021

We thank the editor and reviewers for the thorough examination of our manuscript and for providing positive and helpful feedback. We appreciate the opportunity to address reviewer comments here. Our responses are prefaced by RESPONSE:

Reviewer #1: Review: Protein prediction for trait mapping in diverse populations

Overview: Schubert et al present work on predicting protein abundance in the TOPMed data using genotype information into large-scale GWAS across diverse populations. Their work builds conceptually on the TWAS model, which predicts steady state gene expression (mRNA levels) rather than downstream protein abundance. They investigated the predictive performance and transferability of prediction models across four TOPMed MESA populations (African Americans, Chinese, European, and Hispanic/Latino). Particular attention was paid to the performance of fine-mapping or baseline prediction models across populations, and follow-up with out-of-sample performance in the INTERVAL protein study. Next, the applied their prediction models to GWAS data for 28 phenotypes of diverse populations in the PAGE consortium.

I find this area to be interesting and useful, however I find the results as presented to provide very little insight and lack clear take-aways for downstream decision making. For example, the authors performed a good deal of analyses to quantify protein prediction accuracy, but fail to provide a clear recommendation on which approach to use (e.g., does having more models matter upfront, or having population matched model/GWAS downstream). I appreciate the complexity of multiple analyses across diverse populations complicates matters, but simple meta-analysis tools can simplify the big picture and present results in a consistent manner that help the reader understand what approaches work better on average. Similarly, there are a number of analyses that should have been performed to help place findings in context prior to predictive modeling. Overall, I think the data generated in this manuscript to be interesting and valuable to the broader community, but that a simplified presentation could greatly help with communicating primary findings and informing downstream TWAS/PWAS users.

RESPONSE: Thank you for your helpful recommendations. We have updated our results as described in our responses to your Major Comments below and have added clearer recommendations to our Discussion, lines 322-327:

“Given the improved cross-population prediction of fine-mapped models (S7 Fig, S5 Table, S6 Table) and similar performance to baseline models in PWAS (Fig 5), we recommend using our fine-mapped models in PWAS. We also recommend population-matching in PWAS when protein model training sample sizes are within the same order of magnitude, as in TOPMed MESA, to maximize PWAS discovery, colocalization, and replication.”

I provide more details below.

Major Comments:

1. Given that protein prediction models rely on pQTL signals, it would greatly help if the authors also discussed raw pQTL association (and fine-mapping) results across populations before discussing prediction. Understanding functional enrichment of protein regulatory mechanisms is interesting and crucially unexplored in data at this scale. Including these analyses would help provide context for downstream prediction models and shed light on regulatory mechanisms themselves, prior to their relationship with complex disease risk.

RESPONSE: Thank you for your suggestion. We have added more details about our cis-pQTL mapping to the beginning of the Results (lines 49-60). We added Table 1, which summarizes pQTL counts (FDR < 0.05) and added all pQTL summary statistics to the zenodo repository with the prediction models. We found that effect sizes were enriched near the transcription start site (TSS) for each gene region which mapped to a protein in our sample and that as sample size increased, smaller effect size SNP associations farther from the TSS were discovered (S2 Fig).

2. Similarly, prediction accuracy is inherently tied to heritability. It would be helpful to see how prediction accuracy tracks with in-sample h2g estimates (or out-of-sample INTERVAL R2 with INTERVAL h2g).

RESPONSE: We agree, thank you for the suggestion. We estimated heritability of each protein trait using Bayesian Sparse Linear Mixed Modeling (Zhou et al. 2013), as we have done previously for gene expression traits (Wheeler et al. 2016, Mogil et al. 2018). This analysis is added to the Results (lines 113-119):

“As the heritability of a trait determines the ceiling for genetic prediction performance, we estimated the proportion variance explained (PVE) by SNPs within 1Mb of each protein encoding gene using Basyesian Sparse Linear Mixed Modeling (BSLMM) [35]. Highly heritable proteins (high PVE) were associated with high predictive performance in INTERVAL across populations, despite larger credible sets surrounding the PVE estimates in the smaller populations, i.e., CHN and AFA. (S5 Fig).”

We added a description of our BSLMM analysis to the Methods (lines 501-506):

“We used the software GEMMA to implement BSLMM for each protein aptamer with 100K sampling steps per aptamer. BSLMM estimates the PVE (the proportion of variance in phenotype explained by the additive genetic model, analogous to h2). From the second half of the sampling iterations for each aptamer, we compared the median and the 95% credible sets of the PVE to model performance in INTERVAL.”

3. Can the authors provide some supporting analyses for when genes fail to replicate across populations? It would be interesting to see how avg Fst at a gene/locus tracks with avg cross-pop R2.

RESPONSE: Thank you for the suggestion. We have taken your advice and believe our results make our paper stronger. We added a new figure (now Fig 4) and the following to the Results (lines 145-164):

“When we compared all five TOPMed MESA training populations within each model building strategy, we observed the largest and most significant differences between populations in the baseline models rather than the fine-mapped models (S7 Fig, S5 Table, S6 Table). To test the hypothesis that allele frequency differences between populations influence predictive power, we performed a fixation index (FST) analysis. For each model set, we calculated the (FST) between INTERVAL and the corresponding TOPMed population for SNPs in the predictive model. We then compared the difference in average (FST) between protein models that had a large difference in predictive performance between populations and protein models that had a small difference (Fig 4). We tested multiple thresholds for differences in predictive performance in both fine-mapped and baseline model sets. We found that models which had minimal differences in their performance had significantly smaller differences in average FST than models which had larger differences in performance by Wilcoxon signed-rank test (Fig 4). This effect was observed for multiple thresholds in both baseline and fine-mapped model sets, but was attenuated in fine-mapped sets. Thus, performance differences between populations in the fine-mapped models are less likely due to allele frequency differences. As sample sizes in proteomics studies increase, allowing identification of SNPs with higher PIP values, including trans-acting pQTLs, we anticipate increased cross-population performance benefit from multi-ancestries fine-mapping.”

Reviewer #2: The authors took up an exploratory study that constructed pQTL models using TOPMed MESA cohorts of various ancestries, including African American, Chinese, European, Hispanic/Latino, and cross-population; models were further evaluated and validated using an independent cohort European INTERVAL. For each population-specific cohort or cross-population cohort, the authors have also developed a baseline model (which I believe was inclusive of all sequenced or genotyped variants) and a fine-mapped model. In general, fine-mapped models outperformed baseline models in terms of significant pQTL signals/models. Furthermore, the authors used the constructed models to perform PWAS on the PAGE cohort and replicated in the UKB+ data when the testing trait is available in the replication cohort. The authors successfully identified several known associations, for example, HDL-APOE.

1. In line 86-90, the authors stated that they identified 372 protein aptamers distinct to MESA and not found in GTEx Whole Blood models. Can there be false positives? Are these protein aptamers population-specific or from the cross-population model? It would explain the distinctiveness of these aptamers if these were population specific.

RESPONSE: Thank you for your questions. The cross-validated prediction performance of all models with R2>0.01 is listed in S3 Table along with columns indicating whether or not the gene has a GTEx whole blood or any tissue transcription model (mashr method in Barbeira et al. 2020). We note many proteins (254/372) that do not have a mashr transcript model in GTEx do have a significant protein aptamer model trained in the MESA EUR population, which is the closest ancestry to GTEx, therefore most aptamers are not population-specific. Yes, we agree that some of the models listed in S3 Table may be false positives, which is why we go on to test them in the independent INTERVAL cohort.

2. For table 1, are these all replication of previous findings? Are some of them novel? There was analysis in the result saying that some significant association signals of APOE isoforms went away after adjusted for PAV. Would it be better if this is noted in the table 1 or at least stated in the table 1 legend?

RESPONSE: Thank you for the suggestion. Yes, the APOE associations were no longer significant after adjusting for PAVs. We agree that this should be noted in what is now Table 2 and have added a footer indicating which associations are no longer significant after PAV adjustment.

We also discuss in lines 221-227 that “Three of our protein-trait associations were not found in the original PAGE GWAS, but are still supported by independent GWAS. Increased Haptoglobin, Mixed Type was associated with decreased LDL cholesterol and decreased total cholesterol, both of which are corroborated by GWAS at this locus (Klarin et al. 2018). Increased IL-1Ra was associated with decreased C-reactive protein. SNPs near IL-1Ra associated with C-reactive protein in an independent GWAS (Han et al. 2020). The directions of effect for each protein-phenotype association were consistent between all training populations.”

3. Were there any related samples in MESA? Did the authors adjust for relatedness among samples?

RESPONSE: Yes, we adjusted for cryptic relatedness using PCAIR, as described in the Methods, lines 431-442. No close relatives (1st-2nd degree) were identified. 

Minor suggestions or side questions

1. What the relationships between pQTLs in this study and MESA eQTLs? Is it correlated?

RESPONSE: We agree this would be a useful analysis, but it would be a significant project beyond the scope of this paper due to differences in tissues, timing, samples, and harmonization issues. For example, the protein data come from plasma, while TOPMed MESA has RNA-Seq data in PBMCs, monocytes, and T-cells taken at different exam timepoints. We note there are other ongoing TOPMed proposed papers performing such integrative analyses.

2. Some acronyms, like PIP, were declared more than once.

RESPONSE: Thank you, we have edited our manuscript so acronyms are declared upon first use and not again.

---

## [Decision Letter · Decision Letter 1]

9 Feb 2022

Protein prediction for trait mapping in diverse populations

PONE-D-21-26641R1

Dear Dr. Wheeler,

We’re pleased to inform you that your manuscript has been judged scientifically suitable for publication and will be formally accepted for publication once it meets all outstanding technical requirements.

Kind regards,

Heming Wang, PhD

Academic Editor

PLOS ONE

Additional Editor Comments (optional):

Reviewers' comments:

Reviewer's Responses to Questions

**Comments to the Author**

1. If the authors have adequately addressed your comments raised in a previous round of review and you feel that this manuscript is now acceptable for publication, you may indicate that here to bypass the “Comments to the Author” section, enter your conflict of interest statement in the “Confidential to Editor” section, and submit your "Accept" recommendation.

Reviewer #1: All comments have been addressed

Reviewer #2: All comments have been addressed

2. Is the manuscript technically sound, and do the data support the conclusions?

Reviewer #1: Yes

Reviewer #2: Yes

3. Has the statistical analysis been performed appropriately and rigorously? 

Reviewer #1: Yes

Reviewer #2: Yes

4. Have the authors made all data underlying the findings in their manuscript fully available?

Reviewer #1: Yes

Reviewer #2: Yes

5. Is the manuscript presented in an intelligible fashion and written in standard English?

Reviewer #1: Yes

Reviewer #2: Yes

6. Review Comments to the Author

Reviewer #1: (No Response)

Reviewer #2: (No Response)

7. PLOS authors have the option to publish the peer review history of their article (what does this mean?). If published, this will include your full peer review and any attached files.

Reviewer #1: No

Reviewer #2: No

---

## [Editor Report · Acceptance letter]

14 Feb 2022

PONE-D-21-26641R1 

Protein prediction for trait mapping in diverse populations 

Dear Dr. Wheeler:

I'm pleased to inform you that your manuscript has been deemed suitable for publication in PLOS ONE. Congratulations! Your manuscript is now with our production department. 

Kind regards, 

on behalf of

Dr. Heming Wang 

Academic Editor

PLOS ONE